

# NEIVAv1.0:
# Next-generation Emissions InVentory expansion of Akagi et al. version 1.0

Samiha Binte Shahid[1], Forrest G. Lacey[2,5], Christine Wiedinmyer[3], Robert J. Yokelson[4], Kelley C. Barsanti[1,5]

[1]Department of Chemical & Environmental Engineering, Center for Environmental Research & Technology, University of California-Riverside, Riverside, 92521, United States

[2]Research Applications Laboratory, U. S. National Science Foundation National Center for Atmospheric Research, Boulder, 80301, United States

[3]Cooperative Institute for Research in Environmental Sciences, University of Colorado-Boulder, Boulder, CO, 80309, United States

[4]Department of Chemistry, University of Montana, Missoula, 59812, United States

[5]Atmospheric Chemistry Observations and Modeling Laboratory, U. S. National Science Foundation National Center for Atmospheric Research, Boulder, 80301, United States

*Correspondence to*: Kelley C. Barsanti (barsanti@ucar.edu)

**Abstract**. Accurate representation of fire emissions is critical for modeling the in-plume, near-source, and remote effects of biomass burning (BB) on atmospheric composition, air quality, and climate. In recent years application of advanced instrumentation has significantly improved knowledge of the compounds emitted from fires, which coupled with a large number of recent laboratory and field campaigns, has facilitated the emergence of new emission factor (EF) compilations. The Next-generation Emissions InVentory expansion of Akagi (NEIVA) version 1.0 is one such compilation in which the EFs for 14 globally-relevant fuel and fire types have been updated to include data from recent studies, with a focus on gaseous non-methane organic compounds (NMOC_g). The data are stored in a series of connected tables that facilitate flexible querying from the individual study level to recommended averages of all laboratory and field data by fire type. The querying features are enabled by assignment of unique identifiers to all compounds and constituents, including 1000s of NMOC_g. NEIVA also includes chemical and physical property data and model surrogate assignments for three widely-used chemical mechanisms for each NMOC_g. NEIVA EF datasets are compared with recent publications and other EF compilations at the individual compound level and in the context of overall volatility distributions and hydroxyl reactivity (OHR) estimates. The NMOC_g in NEIVA include ~4-8 times more compounds with improved representation of intermediate volatility organic compounds resulting in much lower overall volatility (lowest volatility bin shifted by as much as three orders of magnitude) and significantly higher OHR (up to 90%) than other compilations. These updates can strongly impact model predictions of the effects of BB on atmospheric composition and chemistry.



## 1. Introduction

The identification, quantification, and model representation of gaseous and particulate compounds emitted from fires are critical for modeling the effects of biomass burning (BB) on air quality and climate. BB occurs under a variety of conditions and involves a range of plant-based fuels, which vary greatly across the world's ecosystems. In the dry forests of the Western US, long-term policies of wildfire suppression and management practices have led to the accumulation of understory fuels in many forests (Collins et al., 2011). This decades-long shift in forest structure, coupled with a warming climate, greatly increases the potential for destructive wildfires (Stephens et al., 2014; North et al., 2015). Land use and climate trends have driven significant changes in BB in other parts of the world as well, with sometimes uncertain effects on air quality and climate (Doerr and Santín, 2016). Some examples include a lengthening of the fire season and increased area burned in boreal forests (de Groot et al., 2013; Jolly et al., 2015), an increase in fire severity and area burned in tropical peatlands (Page and Hooijer, 2016), and a decrease in area burned in sub-Saharan Africa with conversion of savanna to croplands (Andela and van der Werf, 2014; Hickman et al., 2021).

On a global scale, fires emit large amounts of trace gases, including nitrogen oxides ($NO_x$), carbon monoxide (CO), and carbon dioxide ($CO_2$); non-methane organic compounds (NMOCs); and primary (directly emitted) particulate matter (PM). Emission rates and properties of gaseous and particulate compounds are highly variable and largely dependent on fuel characteristics and burn conditions (Guyon et al., 2005; Yokelson et al., 2007; McMeeking et al., 2009; Jolleys et al., 2012; Urbanski, 2014; Liu et al., 2017). During plume dilution directly emitted PM, a large fraction of which is organic (Zhao et al., 2013; Liu et al., 2017), can evaporate reducing the amount of primary organic aerosol (POA), but also adding reactive gases, e.g., semi-volatile NMOCs (Bian et al., 2017; Hodshire et al., 2019). During plume evolution gaseous NMOCs (NMOC_g) may react to form ozone ($O_3$); secondary PM, more commonly referred to as secondary organic aerosol (SOA); and other secondary products that can degrade air quality and endanger human health (Crutzen and Andreae, 1990; Poschl, 2005; McClure and Jaffe, 2018; Buysse et al., 2019; Wei et al., 2023). Model representation of the NMOC_g and the ambient conditions (e.g., light, oxidant, and $NO_x$ levels), are important for accurate predictions of $O_3$, SOA, and other pollutants (Alvarado et al., 2009; Tkacik et al., 2017; Ahern et al., 2019; Hatch et al., 2019; Decker et al., 2019, 2021; Ninneman and Jaffe, 2021; Xu et al., 2021; Fredrickson et al., 2022).

Application of advanced instrumentation has significantly improved estimates of gaseous and particulate compounds emitted from fires in recent years. For example, high-resolution chemical ionization mass spectrometry, CIMS (Stockwell et al., 2015; Koss et al., 2018; Palm et al., 2020), and one- and two-dimensional gas chromatography with time-of-flight mass spectrometry, GC-TOF-MS and GC×GC-TOF-MS (Hatch et al., 2015; Gilman et al., 2015; Hatch et al., 2019; Jen et al., 2019; Liang et al., 2021) have expanded the capacity to measure organic compounds with diverse chemical and physical properties, making it possible to identify and quantify much of the previously-ubiquitous unknown emissions (Christian et al., 2003; Warneke et al., 2011). Laboratory studies that carefully simulated globally-relevant fuels and fire types enabled initial measurements with these new techniques (Stockwell et al., 2014; Hatch et al., 2015; Selimovic et al., 2018) and the development of comprehensive NMOC_g datasets (Koss et al., 2018; Hatch et al., 2017). Incandescence (Schwarz et al., 2006) and photoacoustic (Lewis et al., 2008; Nakayama et al., 2015) techniques for measuring black carbon, BC, have overcome some of the limitations




with older thermal and thermal-optical approaches for measuring elemental carbon, EC (Li et al., 2019). Online aerosol measurements with the Aerodyne aerosol mass spectrometer (AMS), along with offline filter-based measurements, have greatly expanded particulate emissions datasets (Jayarathne et al., 2018; Jen et al., 2019). Application of these and other techniques during field campaigns have led to improved characterization of emissions from specific fuel and fire types, including peat fires in Borneo (Stockwell et al., 2016a; Yokelson et al., 2022); cooking fires, agricultural fires, and garbage burning in Nepal (Stockwell et al., 2016b); and most notably, wildfires and agricultural burns in the US (Liu et al., 2016, 2017; Permar et al., 2021; Gkatzelis et al., 2023; Travis et al., 2023).

The large number of laboratory and field campaigns, and rapid expansion of published BB emissions datasets, has facilitated the emergence of new emission factor (EF) compilations, including Andreae (2019) an update to the 2001 compilation of Andreae and Merlet (2001) and the Smoke Emissions Repository Application, SERA (Prichard et al., 2020) an update to the 2014 Wildland Fire Emissions Database (Lincoln et al., 2014). The Andreae (2019) inventory includes EFs for 121 gas- and particle-phase species or constituents (i.e., total PM); the data are almost entirely from field measurements and include a range of globally-relevant fuel and fire types. The SERA database (Prichard et al., 2020) includes EFs for 276 gas- and particle-phase species or constituents; the focus of the database is North American wildland fuels and both laboratory and field data are included. Similarly to Andreae (2019) the NEIVA (Next-generation Emissions Inventory expansion of Akagi) database described herein includes EFs for globally-relevant fuel and fire types, but in contrast to Andreae (2019) 1000s of compounds and representative laboratory data were selectively included. Similarly to SERA (Prichard et al., 2020) NEIVA is an online, searchable database that includes source data and recommended average EFs across fuel and/or fire types. Additional features unique to NEIVA are summarized below, and detailed in the manuscript Sections 2-5, the Supplementary Information (SI), and on GitHub (https://github.com/NEIVA-BB-Emissions-Inventory).

In v1.0, NEIVA exists as a collection of datasets and Python script files (summarized in Table S1). The datasets include a primary database (multiple data tables) with collected and reformatted data from existing emission inventories and recent laboratory and field campaigns, and a recommended EF dataset (single data table) with EFs averaged across studies and summarized for 14 globally-relevant fuel and fire types. NEIVA also includes a property dataset that links each NMOC_g with a suite of chemical and physical properties using unique identifiers. Because one function of emission inventories in models is to distribute the total gaseous NMOC emitted from fires among the suite of compounds or lumped model species represented in the model, each of the NMOC_g in the NEIVA database has been mapped to SAPRC (Carter, 2010, 2020, 2023a), MOZART-T1(Emmons et al., 2020), and GEOS-Chem (Bey et al., 2001; Carter et al., 2022) model surrogates. Using the Python script files, NEIVA can produce detailed NMOC_g speciation profiles for different fuel and/or fire types, as well as lumped NMOC_g speciation profiles in which individual compounds are mapped to model surrogates. The inclusion of recent laboratory and field data within NEIVA results in significant differences in the molar, mass, and property distributions of NMOC_g as individual compounds and as mapped to model surrogates when compared with existing inventories. The data underlying NEIVA are described in Section 2. The structure and contents of NEIVA are described in Section 3. Evaluation of the data processing steps to generate the datasets within NEIVA and differences between NEIVA and existing EF compilations are



presented in Section 4, including implications of these differences on atmospheric composition and air quality predictions. Examples of querying commands and data products are presented in Section 5. Further details on the processes and procedures used to create the datasets, and additional verification and validation, are presented in the SI. NEIVA can be accessed through the GitHub page: https://github.com/NEIVA-BB-Emissions-Inventory/NEIVAv1.0, which includes

detailed instructions and Jupyter notebooks for querying EF data and adding EF data using the associated script files.

## 2. Data

### 2.1 Legacy data and structure ("NEIVA legacy database")

In 2011, Akagi et al. (2011) published a compilation and assessment of EFs for domestic and open

BB and garbage burning (GB), which included recommended EFs based on literature averages. The overarching aim of the 2011 paper was to compile EF data from numerous field studies of fresh plumes, especially for NMOC_g, that had been published in the ten years since the 2001 Andreae and Merlet (2001) compilation. Some additional useful features in the 2011 paper include: 1) discussions of BB terminology, combustion chemistry, photochemistry in young plumes,

tracers, and other relevant topics; 2) a table of published measurements of fuel consumption per unit area for major types of open burning; 3) examples of scaling to global estimates; 4) methods to estimate unmeasured species; and 5) updated EFs for some species (notably formic acid and glycolaldehyde) based on new infrared reference data. In addition, as relevant to this work, Akagi et al. (2011) expanded from 7 to 14 representative fuel types, included more species, and provided

estimated EFs for the sum of unknown species.

In Akagi et al. (2011) the selected EFs for each species in each study were explicitly shown in 14 supplemental tables organized by fuel or fire type. Also shown in the supplemental tables was a reasonably-simple and transparent averaging scheme (detailed and justified in the Akagi et al.

manuscript) designed to make the literature averages representative. Between 2014 and 2015, some of the SI tables were updated online (Wiedinmyer et al., 2014), specifically temperate forest and chaparral in 2014 and savanna in 2015. In these updates, compounds were listed in mass order while still providing common names, to solve the problem of multiple common names and to enhance the ability to quickly locate specific compounds. NEIVA builds on the Akagi et al. (2011)

EF data and their updates through 2015. These data are referred to as the "legacy database" in NEIVA and are included as a series of 14 tables (listed in Table S2). Each table includes the data as presented by Akagi et al. (2011) (see Table S3), as well as unique identifiers assigned in this work to link datasets within NEIVA. Since 2015, lists of new papers with useful EFs were posted online and organized by the original 14 fuel and fire types in Akagi et al. (2011), and they included

brief comments on paper content, while contemplating how best to progress given the frequent appearance of new data and the expanding number of compounds measured. The next section gives brief updates on the progress, or lack thereof, for each of these original 14 fuel and fire types.

### 2.2 New data and structure ("NEIVA raw database")

Based largely on the lists posted online since 2015, data from a total of 30 publications associated

with 12 of the 14 fuel and fire types have been compiled and are referred to here as the "raw





database". Data from these publications were included in NEIVA as a series of 30 tables (listed in Table S5). The publications and data are introduced under the relevant fuel or fire categories below. One category, peatland, has been removed from the legacy fuel categories (see S2) and one category under domestic BB has been added (see S2, Table S7). These revisions and any other major changes to the categories are described in further detail below. The new data include field and laboratory data from single-institution studies to multi-institution campaigns, including the 4th Fire Laboratory at Missoula Experiment, FLAME-4 (Stockwell et al., 2014); Western Wildfire Experiment for Cloud chemistry, Aerosol Absorption, and Nitrogen, WE-CAN (Juncosa Calahorrano et al., 2021); Fire Influence on Regional to Global Environments and Air Quality, FIREX laboratory and FIREX-AQ field (Warneke et al., 2023); and Nepal Ambient Monitoring and Source testing Experiment, NAMaSTE (Jayarathne et al., 2018).

### 2.2.1 Savanna fires

The Akagi et al. (2011) savanna fire table was updated in February 2015 with extensive PTR-ToF-MS data from FLAME-4. There have been no large-scale field campaigns measuring fire EFs in tropical savannas since SAFARI 2000. However, Desservettaz et al. (2017) reported new BB EFs for several gaseous compounds and particulate constituents measured during a field study in Australian savannas and the data were included here. In addition, Travis et al. (2023) reported EFs for gaseous compounds and particulate constituents from prescribed burns of grasslands in the midwestern US that were included here.

### 2.2.2 Boreal forest

EFs were included here for over 190 gas- and particle-phase compounds or constituents reported by Hayden et al. (2022) based on airborne sampling of a smoldering boreal forest fire. In addition, black spruce from Alaska was burned during the FLAME-4 laboratory studies and the associated EFs reported by Stockwell et al. (2015) and Hatch et al. (2015) were included here (see Table S6 for mapping of individual fuels to the 14 representative fuel and fire types).

### 2.2.3 Tropical forest

Several new EFs were included for particulate compounds or constituents reported by Hodgson et al. (2018) for evergreen tropical forest and cerrado (seasonally dry tropical forest, aka "monsoon forest") measured during the 2018 SAMMBA campaign.

### 2.2.4 Temperate forest

The Akagi et al. (2011) temperate forest table was updated in May of 2014. Since that update, several relevant papers have been published and the EF data were included here. Data were included for wildfires and prescribed burns (tagged accordingly in the datasets). Liu et al. (2017) reported EFs for many gas- and particle-phase species and constituents for western US wildfires from the 2013 SEAC[4]RS and BBOP field campaigns. Permar et al. (2021) reported EFs for 161 NMOC_g and particle-phase constituents largely from wildfires sampled in the 2018 WE-CAN field campaign. Gkatzelis et al. (2023) reported EFs for 98 NMOC_g and four particulate constituents (nitrate; ammonium; black carbon, BC; organic aerosol, OA) also largely from wildfires sampled during the 2019 FIREX-AQ campaign. Travis et al. (2023) reported EFs for 148 NMOC_g and ten particulate constituents (PM ≤ 1 microns, PM$_1$; BC; organic carbon, OC; OA;



ammonium chloride; potassium; nitrate; sulphate) for prescribed burns (slash, pile, and Blackwater River State Forest understory) of temperate forest fuels measured in the midwestern US during FIREX-AQ. Müller et al. (2016) published NMOC_g EFs for a small prescribed fire in the SE US. The old nephelometer-based temperate forest prescribed fire $PM_{2.5}$ (PM ≤ 2.5 microns) EFs from Burling et al. (2011) were replaced with new $PM_1$ EFs for the same fires based on AMS data from May et al. (2014). Laboratory-based wildfire simulations were conducted during FLAME-4 (Stockwell et al., 2014) and FIREX (Selimovic et al., 2018), resulting in new EFs for gas- and particle-phase species and constituents (Stockwell et al., 2015; Hatch et al., 2015, 2017; Koss et al., 2018; Selimovic et al., 2018). EF data for relevant fuels from FLAME-4, ponderosa pine and juniper, and most of the FIREX laboratory burns were included here, as listed in Table S6.

**2.2.5 Peat**

Peat is often thought of as a single fuel that burns by smoldering in the field and therefore, in theory, should be easy to burn representatively in a laboratory (neglecting the challenge of obtaining international samples). However, in reality the type of peat varies with depth for undisturbed sites and in more complex ways for disturbed sites (Stockwell et al., 2016a), which translates into additional uncertainties for laboratory-based emissions measurements. Artificially low % C values reported in the literature for some peat samples suggests that such samples contained significant amounts of mineral soil and thus resulted in low bias for associated EFs. Further, peat ignition can be difficult, and aggressive ignition with a propane torch can lead to unrepresentative flaming. Such cases have been identified by high modified combustion efficiency (MCE) values, $NO_x$, and/or high acetylene ($C_2H_2$) emissions (e.g., $C_2H_2/C_2H_4 > 1$) and have been omitted here. In field studies, random sampling of real peat fires should return representative values, but interference from the emissions from other fuels can be difficult to avoid and potential storage artifacts for off-line analyses also may be unavoidable if shipping delays are encountered. After carefully screening for all these effects, some excellent new data emerged.

Four papers presented new field measurements of "pure" tropical peat fires. Jayarathne et al. (2018) reported comprehensive filter-based EFs ($PM_{2.5}$, EC, OC, numerous organic compounds, metals, etc.) from measurements obtained during the 2015 El Niño in Borneo. Stockwell et al. (2016a) reported EFs for ~100 gases, BC, brown carbon (BrC), and aerosol optical properties from the same study. Smith et al. (2018) measured trace gas EFs on authentic peat fires in Malaysia and Roulston et al. (2018) measured $PM_{2.5}$ EFs on peat fires also in Malaysia. Data from all four publications were included here.

Laboratory studies of peat have provided much more detail than has been possible in field studies to date. Peat-fire EFs from both the FLAME-4 and FIREX laboratory studies were included here. As part of FLAME-4, Stockwell et al. (2015) reported EFs for an extensive list of gas-phase species from two samples each of temperate, boreal, and tropical peat based on PTR-TOF-MS and FTIR measurements. Also as part of FLAME-4, Hatch et al. (2015) used GC×GC-TOF-MS to add EF data for alkanes and other species not detected by PTR-MS or FTIR. They also speciated numerous isomers at exact masses where MS sees a single peak. This groundbreaking application of GC×GC led to EFs for > 600 NMOC_g for an Indonesian peat sample. Aerosol optical properties and $PM_{2.5}$ EFs for peat from FLAME-4 reported by Jayarathne et al. (2014) an Pokhrel et al. (2016) were included here. More recently, the FIREX laboratory experiments resulted in EFs for an extensive list of gas-phase species for an Indonesian peat sample based on measurements described in



Selimovic et al. (2018) and Koss et al. (2018). The EFs in the latter study were recalculated here using the actual % C value for the peat provided in Selimovic et al. (2018). Watson et al. (2019) reported laboratory-based EFs for several trace gases for peat samples from the boreal through tropical zones, which were included here, with the exception of EFs for nitrous oxide ($N_2O$) due to the difficulty of decoupling $N_2O$ from high levels of CO and $CO_2$ by infrared spectroscopy.

### 2.2.6 Chaparral

The Akagi et al. (2011) chaparral table was updated in May 2014. Since then, FIREX provided comprehensive EFs for gases reported by Koss et al. (2018) and Selimovic et al. (2018). In these laboratory studies, chaparral was represented by burning two dominant shrub species: manzanita and chamise. The EFs for NMOC_g and particulate constituents reported by Travis et al. (2023) for prescribed burns of shrublands in the midwestern US also were included here, making this category representative of shrub types beyond chaparral.

### 2.2.7 Domestic biomass burning

Domestic (household) biofuel use includes many fuels and burning options that are primarily for cooking, but also heating. Akagi et al. (2011) presented study-level results (in their SI) and "global averages" for five domestic biofuel activities: 1) open cooking (e.g., three stone fires with wood fuel only, believed to be the most common type of domestic biofuel use), 2) wood cooking with improved stoves (including "rocket type" stoves only, which were believed to be the most common improved stove), 3) charcoal making, 4) charcoal burning (open or in improved stoves), and 5) dung burning (open or in improved stoves). Since 2011, many new improved stove designs have been developed and characterized, many new EFs have been measured, and results for mixed fuel cooking fires (e.g., wood and dung) have been published. To capture the new results slightly revised categories were established as follows (see Table S7): 1) open cooking (three stone and wood), 2) cookstove (traditional and modern), and 3) dung burning (w/ and w/o wood, traditional and modern). Since there is no systematic approach for grouping fuels and stoves in the literature, the above approach has been adopted here while tagging data appropriately in the raw database to facilitate custom selection of relevant data by users. The charcoal making and charcoal burning categories were retained.

**Open cooking:** The open cooking fire type includes all open wood cooking (i.e., three stone). Data from three new publications on various types of open cooking were included here. EFs for gases and aerosol optical properties for open cooking with wood were measured in-situ in Nepal as part of the NAMaSTE campaign and reported by Stockwell et al. (2016b). Gravimetric $PM_{2.5}$ data and chemical speciation of PM from the same study were reported by Jayarathne et al. (2018). EFs for $CO_2$, CO, and $PM_{2.5}$ were measured for a variety of traditional and improved stoves in Ghana by Coffey et al. (2017) and the data for three stone wood burning were included here. Laboratory-based EFs were included here from the carefully-simulated open cooking during FLAME-4, with several wood species commonly used in Mexico, reported by Stockwell et al. (2015).

**Cookstoves:** Akagi et al. (2011) limited improved stove data to rocket stoves burning wood, but in NEIVA additional advanced stove types and fuels were included in the cookstove category. Stockwell et al. (2015, 2016b), Jayarathne et al. (2018), and Fleming et al. (2018) reported data



300 for many types of advanced stoves that were included here. For a subset of the same sources in Stockwell et al. (2015, 2016b) and Jayarathne et al. (2018), Goetz et al. (2018) reported EFs for OA, BC, sulfate, nitrate, chloride, ammonium, and polycyclic aromatic hydrocarbons (PAHs) that were included here. EFs for $CO_2$, CO, and $PM_{2.5}$ for improved stoves reported by Coffey et al. (2017) were included here.

305 **Dung burning:** Data from several new studies with EFs for open dung burning, dung burning in stoves, and mixed dung/wood burning have been reported and were included here. Stockwell et al. (2016b), Jayarathne et al. (2018), Goetz et al. (2018), and Fleming et al. (2018) reported data from studies in Nepal and India. In addition, data were included from the open dung burning sampled in detail during the FIREX laboratory experiments as reported by Koss et al. (2018) and Selimovic 310 et al. (2018).

**Charcoal making:** Literature searches suggest there are no new laboratory- or field-based EFs for charcoal making since Akagi et al. (2011) and thus this remains the least-characterized globally-relevant major fuel type.
315

**Charcoal burning:** Stockwell et al. (2016b) and Jayarathne et al. (2018) reported data for charcoal burning in the Nepal study and the reported EFs were included here. EFs for $CO_2$, CO, and $PM_{2.5}$ for charcoal burning reported by Coffey et al. (2017) also were included.

320 ### 2.2.8 Pasture maintenance

Literature searches suggest there are no new laboratory- or field-based EFs for pasture maintenance fires since Akagi et al. (2011).

### 2.2.9 Crop residue

Akagi et al. (2011) highlighted that the NMOC_g EFs from pile burning of crop residue, which is 325 associated with manual harvest and promotes smoldering, are much higher than those for burning residue loose in the field, which is associated with mechanized agriculture and promotes flaming. More recently, Lasko and Vadrevu (2018) estimated the relative amount of these two burning practices in Vietnam. In addition to the inclusion of new data, the Akagi et al. (2011) EFs were updated here to represent the evolving literature average % C. Following Stockwell et al. (2016a, 330 b), the Mexican "loose in field" crop residue EFs from Yokelson et al. (2011) used in Akagi et al. (2011) and Andreae (2019), were normalized to lower fuel % C values (40 %) by multiplying the original Yokelson et al. (2011) values by 0.8.

Regarding new data, field measurements of loose and piled crop residue fires were carried out in 335 Nepal with EFs for gases and aerosol optical properties reported by Stockwell et al. (2016b). EFs for PM constituents reported by Goetz et al. (2018) and EFs from filter-based $PM_{2.5}$ analyses reported by Jayarathne et al. (2018) were included here. Holder et al. (2017) used several platforms to measure emissions from burning residue in wheat and bluegrass fields in the NW US; the reported EFs from the individual observations and averaged across platforms were included here. 340 Also included were the EFs from field measurements of crop residue fires in the SE US made on the NASA-DC-8, from burning rice straw loose in the field, as part of SEAC[4]RS and reported by





Liu et al. (2016); and the EFs from field measurements of crop residue fires also in the midwestern US, mase as part of FIREX-AQ and reported by Travis et al. (2023). During FLAME-4, numerous types of crop residue were burned in the laboratory, both in piles and loose. The EFs for an
extensive selection of gases and residue types reported by Stockwell et al. (2015) and the rice straw emissions reported by Hatch et al. (2015, 2017) were included here (see Table S6). Rice straw EFs measured during a FIREX laboratory pile-burning simulation also were included (Koss et al., 2018; Selimovic et al., 2018; Gkatzelis et al., 2023; Travis et al., 2023).

### 2.2.10 Garbage burning

The Akagi et al. (2011) recommended EFs for garbage burning (GB) were based almost entirely on one field campaign in Mexico (Christian et al., 2010). These data were incorporated into a global GB inventory by Wiedinmyer et al. (2014). New EFs for mixed garbage fires in Nepal for gases and aerosol optical properties reported by Stockwell et al. (2016b); gravimetric $PM_{2.5}$, EC, OC, and chemical speciation reported by Jayarathne et al. (2018); and size distributions and a full
suite of AMS species (OA, OC, ammonium, sulfate, chloride, and nitrate) reported by Goetz et al. (2018) were included here. In addition, laboratory-based GB EF data from Yokelson et al. (2013) and FLAME-4 reported by Stockwell et al. (2015) were included.

## 3. NEIVA structure and contents

### 3.1 Overview

A schematic of NEIVA is shown in Figure 1. NEIVAv1.0 is a collection of linked data tables. Groups of related tables are organized as a single database and include the legacy database and raw database described above in Sections 2.1 and 2.2, respectively, and the primary database described below in Section 3.2. Collections of related data tables are referred to as databases, while single data tables are referred to as datasets. Datasets in NEIVA include the integrated EF,
processed EF, recommended EF, and chemical property and surrogate ('property_surrogate') datasets, which collectively comprise the output database and are described below in Section 3.3. Each of the databases and datasets are listed in Table 1. In this section, the structure and contents of the primary database and of the integrated EF, processed EF, recommended EF, and chemical property and surrogate datasets are further described, as well as the formatting and data processing
steps that were performed to create each of the data tables. All of the datasets can be accessed through GitHub and the recommended EF dataset is also provided here as a Supplemental Table.

All of the compounds or constituents in the NEIVA database were assigned one of the following pollutant categories: inorganic gas, methane, gaseous non-methane organic compound
(NMOC_g), particulate non-methane organic compound (NMOC_p), or particulate matter (PM). The PM was further differentiated as "size" (e.g., $PM_1$, $PM_{2.5}$, $PM_{2.5}^*$ ($PM_{1-5}$), $PM_{10}$), "organic" (e.g., OA, OC), "elemental" (e.g., EC, BC), "ion" (e.g., Na), "metal" (e.g., lead), and "optical" (e.g., absorption/backscattering coefficients at specific wavelengths). All tables in the legacy, raw, and primary databases include the following columns: molar mass (mm), chemical
formula (formula), compound (compound name), pollutant category, EF, and unique ID for each compound or constituent. Additional information from the source publications was retained in the databases as described in S1. In the EF datasets, each row in a table represents a chemical



compound or constituent, and the columns represent attributes of that compound or constituent, primarily EFs. The algorithm and approach for assigning the unique IDs are described in S1. The unique IDs are one of the critical features for creating and linking the datasets.



**Figure 1: Schematic of NEIVA. The use of "contextual" here (data processing phase 1) refers to information to provide additional context for EF data including: measurement location (lab/field), fuel type, modified combustion efficiency (MCE), and publication identifiers (e.g., DOI, year).**



| Table 1: Description of the databases (multiple related data tables) and datasets (single data tables) that comprise NEIVA. | |
| --- | --- |
| **Data Storage Name** | **Description** |
| Legacy database (ldb) | The Akagi et al. (2011) supplemental data, including 2014 and 2015 updates, are stored as tables in this repository. There are 14 tables, one for each fuel or fire type. All compounds and constituents were assigned a unique id. No data processing performed. |
| Raw database (rdb) | Data from selected publications (2015 or later) are stored as tables in this repository. There are 30 tables in this database: one for each of the publications added since Akagi et al.(2011). All compounds and constituents were assigned a unique id. No data processing performed. |
| Primary database (pdb) | Data from the legacy and raw database tables were reformatted to achieve a consistent structure and combined with some data processing as described in the manuscript and S2, namely updates to the % C for some reported fuels. The resultant 44 tables are stored in this repository. |
| NEIVA output database (odb) | Four datasets are stored in the NEIVA output database:<br><br>Integrated EFs: EF data aggregated in the primary database were merged and stored in this single dataset for all fuel and fire types. The process for merging EFs is described in the manuscript and S3.<br><br>Processed EFs: Additional data processing steps were performed on the integrated EF dataset prior to calculating recommended EFs, as described in S4. Laboratory data were adjusted to represent known differences in combustion conditions between laboratory and field studies. Groups of isomeric compounds were resolved and assigned fractional contributions when possible.<br><br>Recommended EFs: The arithmetic means of the processed EFs for each compound or constituent in each of the 14 representative fuel or fire types are stored in this single dataset. Prior to averaging, $NO_x$ EFs were converted to "$NO_x$ as NO" EFs if NO and $NO_2$ data were available (see S5).<br><br>Property_Surrogate: For each of the gaseous organic compounds in these datasets, chemical and physical property data, as well as model surrogate assignments for specific chemical mechanisms, are stored in this single dataset (see S6). |
| Backend database (bdb) | Tables that are used in the Python scripts for data processing, listed and described in S8, are stored in this database. The tables in the backend database were used to create the output datasets but are not necessary for users to access the EF data. |

### 3.2 Primary database

Prior to combining the legacy and raw databases to form the primary database, several formatting and data processing steps were performed. The data processing steps on the legacy database included removing peatland and the estimated temperate forest EFs that were included in Akagi et al. (2011) (and were retained in the legacy database), removing EFs for unknown proton ion transfer (PIT) masses for temperate forest and chaparral, combining isomers, and calculating a study average for any studies that reported multiple EFs for a given fuel or fire type. From the raw
database, the EFs reported by Koss et al. (2018) were recalculated to reflect measured % C as reported by Selimovic et al. (2018). Further detail on these and additional data processing steps is provided in S2. The resultant primary database is comprised of 44 tables (listed in Table S8). The tables represent the Akagi et al. (2011) EF data separated by fuel or fire type (14 tables) and the EF data from publications since 2015 (30 tables). For publications that include data for a single
fuel or fire type, a fuel designation abbreviation precedes the table name, and otherwise for publications that include data for multiple fuel or fire types, the table name only reflects the source publication (see S2, Table S8 for examples).





### 3.3 Output database

Four datasets are stored in the output database, each of which are described in further detail below. These include the integrated EF dataset, the processed EF dataset, the recommended EF dataset, and the chemical property and chemical mechanism assignment (model surrogate) dataset.

#### 3.3.1 Integrated EF dataset

The aggregated EF data from the tables in the primary database were merged across all studies into a single EF dataset. An algorithm was developed to merge data from individual studies across
tables in the primary database. The algorithm uses a multistep process to group compounds across datasets, determine whether the compounds are the same or different, and then append each compound to the integrated dataset as a new row (indicating a new compound) and each EF as a new column (indicating a new EF). In this dataset, EFs are available for a total of 1311 compounds or constituents with up to 263 measurements (i.e., EFs) study-averaged for individual fuel types
from the primary database. Details on the integration algorithm are provided in S3 and illustrated in Tables S11-S13.

#### 3.3.2 Processed EF dataset

Following integration, the EF data from laboratory studies were corrected to account for known differences between laboratory and field combustion conditions. The results of this correction are
presented and discussed in 4.1, with further detail on the correction methods presented in S4. In addition, to minimize over- or under-counting of individual NMOC_g and to increase the number of measured EFs per individual gaseous NMOC (and thus the statistical robustness), where applicable speciated EF data were used to assign fractional contributions to EFs representing groups of compounds that could not be differentiated using the published method of detection. For
example, because methyl vinyl ketone (MVK) and methacrolein have the same molar mass they are not differentiable by PTR-MS, and thus are often reported as a sum (MVK + methacrolein). For fuel and fire types in which EFs were reported for MVK and methacrolein as a sum and as individual compounds (e.g., using GC×GC-TOF-MS, GC-PTR-MS), the relative EFs of the individual compounds were used to assign fractional contributions to the summed EF, resulting in
two (or more) EFs for MVK and for methacrolein, and no EF for MVK + methacrolein in the processed data set. The results of assigning fractional contributions are presented and discussed in 4.2, with further detail on the fractional assignment presented in S4.

#### 3.3.3 Recommended EF dataset

The arithmetic means of the EFs in the processed dataset were calculated to obtain a single recommended EF for each compound or constituent in each of the 14 fuel or fire types, with equal weighting of the laboratory-adjusted and field EF data. These recommended EFs, along with standard deviation (1σ), data count (number of studies), and emission ratios (ERs) to CO were stored in the recommended EF dataset and are available in the Supplemental Table. A subset of
the Supplemental Table is represented in Tables 2 (EFs) and Table 3 (ERs) below; ERs may be particularly useful in modeling studies where emissions are not explicitly defined. Prior to calculating the recommended EF for each compound or constituent, one additional data processing



step was performed: for studies in which EFs for NO, $NO_2$, and $NO_x$ were reported, $NO_x$ EFs were converted to "$NO_x$ as NO" EFs (see S5). In the recommended EF dataset, for savanna fires, the EF
for OA is greater than the EF for $PM_{2.5}$. The OA represents a single value reported by Travis et al. (2023). In their paper, the EFs for $PM_1$, OA, and OC are self-consistent and reasonable. When averaged here with the other data, because there is only one EF for OA and many EFs for $PM_{2.5}$, the Travis et al. (2023) data disproportionately affect the $EF_{OA}$. The Travis et al. (2023) data were not considered outliers but representative of the natural variability of fuel and fire conditions, and
thus the data were not removed.

**Table 2**: Recommended EFs (g/kg) for selected compounds and constituents.

| | Savanna | Boreal Forest | Tropical Forest | Temperate Forest | Peat | Chaparral | Crop Residue | Garbage Burning |
|---|---|---|---|---|---|---|---|---|
| Carbon dioxide ($CO_2$) | $1.640\times10^3$ | $1.610\times10^3$ | $1.625\times10^3$ | $1.581\times10^3$ | $1.572\times10^3$ | $1.649\times10^3$ | $1.441\times10^3$ | $1.502\times10^3$ |
| Carbon monoxide (CO) | $8.10\times10^1$ | $1.00\times10^2$ | $1.11\times10^2$ | $9.60\times10^1$ | $2.25\times10^2$ | $6.66\times10^1$ | $5.75\times10^1$ | $5.20\times10^1$ |
| Methane ($CH_4$) | $2.83\times10^0$ | $4.78\times10$ | $4.68\times10^0$ | $4.74\times10^0$ | $1.11\times10^1$ | $2.57\times10^0$ | $2.14\times10^0$ | $3.06\times10^0$ |
| Nitric oxide (NO) | $1.76\times10^0$ | $9.16\times10^{-1}$ | $9.00\times10^{-1}$ | $7.85\times10^{-1}$ | $3.21\times10^{-1}$ | $1.15\times10^0$ | $9.62\times10^{-1}$ | 8.10E-01 |
| Nitrogen oxides ($NO_x$ as NO) | $3.40\times10^0$ | $1.21\times10^0$ | $2.55\times10^0$ | $1.65\times10^0$ | $9.27\times10^{-1}$ | $2.42\times10^0$ | $2.05\times10^0$ | $2.31\times10^0$ |
| Nitrogen dioxide ($NO_2$) | $2.60\times10^0$ | $9.22\times10^{-1}$ | $3.55\times10^0$ | $1.40\times10^0$ | $5.43\times10^{-1}$ | $1.02\times10^0$ | $1.96\times10^0$ | $2.34\times10^0$ |
| Nitrous oxide ($N_2O$) | $1.40\times10^{-1}$ | $2.05\times10^{-1}$ | | $1.55\times10^{-1}$ | | $2.50\times10^{-1}$ | | |
| Nitrous acid (HONO) | $4.99\times10^{-1}$ | $2.55\times10^{-1}$ | $1.18\times10^0$ | $3.78\times10^{-1}$ | $2.22\times10^{-1}$ | $5.52\times10^{-1}$ | $3.53\times10^{-1}$ | $2.51\times10^{-1}$ |
| Sulphur dioxide ($SO_2$) | $9.44\times10^{-1}$ | $5.64\times10^{-1}$ | $4.03\times10^{-1}$ | $9.50\times10^{-1}$ | $2.06\times10^0$ | $5.53\times10^{-1}$ | $1.25\times10^0$ | $7.05\times10^{-1}$ |
| Isocyanic acid (CHNO) | $1.05\times10^0$ | $8.30\times10^{-2}$ | | $4.05\times10^{-1}$ | $5.74\times10^{-1}$ | $3.02\times10^{-1}$ | $4.69\times10^{-1}$ | $1.29\times10^{-1}$ |
| Ammonia ($NH_3$) | $6.59\times10^{-1}$ | $1.47\times10^0$ | $1.33\times10^0$ | $1.06\times10^0$ | $6.15\times10^0$ | $9.09\times10^{-1}$ | $9.68\times10^{-1}$ | $6.88\times10^{-1}$ |
| Gaseous Non-Methane Organic Compounds (NMOC_g) | $3.73\times10^1$ | $4.05\times10^1$ | $2.53\times10^1$ | $4.25\times10^1$ | $7.37\times10^1$ | $2.17\times10^1$ | $3.81\times10^1$ | $3.36\times10^1$ |
| $PM_{2.5}^{*(a)}$ | $1.6\times10^1$ | $1.28\times10^1$ | $9.11\times10^0$ | $1.79\times10^1$ | $2.48\times10^1$ | $1.51\times10^1$ | $1.27\times10^1$ | $9.68\times10^0$ |
| OA | $2.73\times10^1$ (b) | $6.60\times10^0$ | | $1.71\times10^1$ | | $1.08\times10^1$ | $1.12\times10^1$ | $7.36\times10^0$ |
| OC | $6.49\times10^0$ | | $3.99\times10^0$ | $1.04\times10^1$ | $1.32\times10^1$ | $1.08\times10^1$ | $9.47\times10^0$ | $5.47\times10^0$ |
| BC | $3.50\times10^{-1}$ | $1.30\times10^{-1}$ | $3.44\times10^{-1}$ | $4.35\times10^{-1}$ | $1.60\times10^{-2}$ | $6.24\times10^{-1}$ | $4.46\times10^{-1}$ | $1.98\times10^0$ |
| EC | | | | | $4.32\times10^{-1}$ | | $4.99\times10^{-1}$ | $1.92\times10^{-1}$ |

(a)$PM_{2.5}^{*}$ includes $PM_1$-$PM_5$. (b) OA is a single value from Travis et al. (2023) that is less than $PM_1$ from the same study.






| Table 3: Recommended ERs (ppb/ppm CO) to CO for selected compounds and constituents. | | | | | | | | |
|---|---|---|---|---|---|---|---|---|
| | Savanna | Boreal Forest | Tropical Forest | Temperate Forest | Peat | Chaparral | Crop Residue | Garbage Burning |
| Carbon dioxide ($CO_2$) | $1.289\times10^4$ | $1.020\times10^4$ | $9.335\times10^4$ | $1.049\times10^4$ | $4.447\times10^3$ | $1.576\times10^4$ | $1.594\times10^4$ | $1.837\times10^4$ |
| Methane (CH4) | $6.10\times10^1$ | $8.31\times10^1$ | $7.38\times10^1$ | $8.63\times10^1$ | $8.61\times10^1$ | $6.74\times10^1$ | $6.50\times10^1$ | $1.03\times10^2$ |
| Nitric oxide (NO) | $2.02\times10^1$ | $8.50\times10^0$ | $7.58\times10^0$ | $7.63\times10^0$ | $1.33\times10^0$ | $1.61\times10^1$ | $1.56\times10^1$ | $1.45\times10^1$ |
| Nitrogen oxides ($NO_x$ as NO) | $3.92\times10^1$ | $1.12\times10^1$ | $2.15\times10^1$ | $1.61\times10^1$ | $3.85\times10^0$ | $3.39\times10^1$ | $3.33\times10^1$ | $4.14\times10^1$ |
| Nitrogen dioxide (NO2) | $1.95\times10^1$ | $5.59\times10^0$ | $1.95\times10^1$ | $8.91\times10^0$ | $1.47\times10^0$ | $9.35\times10^0$ | $2.07\times10^1$ | $2.74\times10^1$ |
| Nitrous oxide ($N_2O$) | $1.10\times10^0$ | $1.30\times10^0$ | | $1.03\times10^0$ | | $2.39\times10^0$ | | |
| Nitrous acid (HONO) | $3.67\times10^0$ | $1.51\times10^0$ | $6.35\times10^0$ | $2.35\times10^0$ | $5.88\times10^{-1}$ | $4.94\times10^0$ | $3.65\times10^0$ | $2.87\times10^0$ |
| Sulfur dioxide (SO2) | $5.10\times10^0$ | $2.45\times10^0$ | $1.59\times10^0$ | $4.33\times10^0$ | $4.01\times10^0$ | $3.63\times10^0$ | $9.47\times10^0$ | $5.92\times10^0$ |
| Isocyanic acid (CHNO) | $8.45\times10^0$ | $5.38\times10^{-1}$ | | $2.75\times10^0$ | $1.66\times10^0$ | $2.96\times10^0$ | $5.31\times10^0$ | $1.61\times10^0$ |
| Ammonia ($NH_3$) | $1.34\times10^1$ | $2.41\times10^1$ | $1.97\times10^1$ | $1.82\times10^1$ | $4.51\times10^1$ | $2.25\times10^1$ | $2.77\times10^1$ | $2.18\times10^1$ |
| Gaseous Non-Methane Organic Compounds (NMOC_g) | $2.31\times10^2$ | $1.99\times10^2$ | $1.35\times10^2$ | $2.04\times10^2$ | $1.67\times10^2$ | $1.66\times10^2$ | $3.10\times10^2$ | $3.45\times10^2$ |
| **ERs (g/g CO)** | | | | | | | | |
| $PM_{2.5}$*[a] | $2.17\times10^{-1}$ | $1.27\times10^{-1}$ | $8.22\times10^{-2}$ | $1.85\times10^{-1}$ | $1.10\times10^{-1}$ | $2.25\times10^{-1}$ | $2.21\times10^{-1}$ | $1.86\times10^{-1}$ |
| OA | $3.38\times10^{-1}$ | $6.57\times10^{-2}$ | | $1.78\times10^{-1}$ | | $1.62\times10^{-1}$ | $1.95\times10^{-1}$ | $1.41\times10^{-1}$ |
| OC | $8.01\times10^{-2}$ | | $3.60\times10^{-2}$ | $1.09\times10^{-1}$ | $5.86\times10^{-2}$ | $1.63\times10^{-1}$ | $1.65\times10^{-1}$ | $1.05\times10^{-1}$ |
| BC | $4.32\times10^{-3}$ | $1.29\times10^{-3}$ | $3.10\times10^{-3}$ | $4.54\times10^{-3}$ | $7.13\times10^{-5}$ | $9.36\times10^{-3}$ | $7.75\times10^{-3}$ | $3.80\times10^{-2}$ |
| EC | | | | | $1.92\times10^{-3}$ | | $8.67\times10^{-3}$ | $3.68\times10^{-3}$ |

(a)$PM_{2.5}$* includes $PM_1$-$PM_5$.



### 3.3.4 Chemical property and model surrogate dataset

In many model applications, it is impractical to represent hundreds of individual organic compounds and thus lumping of compounds is often required. In gas-phase chemical mechanisms, it is typical to lump organic compounds based on their reaction rate constant with OH ($k_{OH}$) and the oxidation products that they form. Groups of compounds may be represented by individual compounds or by model surrogates. To facilitate the use of the comprehensive EF data for NMOC_g included in NEIVA, individual NMOC_g were mapped to model surrogates for the

common gas-phase chemical mechanisms SAPRC-07/-07T/-18/-22 (Carter, 2010, 2020, 2023a), MOZART-T1(Emmons et al., 2020), and GEOS-Chem (Bey et al., 2001; Carter et al., 2022). The methods for assigning the model surrogates and sources for the property data are described in detail in S6. Briefly, compounds were first assigned to the SAPRC and MOZART-T1 mechanisms using the SAPRC Mechanism Generation (MechGen) System web interface (Carter, 2019) and the

SAPRC model species assignment database 'SpecDB'(Carter, 2023b). The SAPRC and MOZART-T1 assignments were then used to determine the GEOS-Chem assignments (see Tables S18-S21), with additional reference to Hutzell et al. (2012), Li et al. (2014), and Carter et al. (2022). The model surrogate assignments are provided in a property dataset (see Table S22) that also includes oxidation rate constants with OH, $O_3$, and $NO_3$ ($cm^3$/molecule-s); vapor pressures

(mm Hg); saturation vapor concentration ($C^*$, μg/$m^3$); Henry's Law constants (atm-$m^3$/mole); O:C ratio; and average carbon oxidation state (Pence and Williams, 2010; NIST Chemistry WebBook, 2022; US EPA, 2023; Kim et al., 2023; ChemSpider, 2024) linked to individual NMOCs by the unique ID.

### 4. Evaluation

### 4.1 Adjustment of Laboratory-Based Emission Factors and Integration of Laboratory and Field Data

Representative laboratory-based EFs were selectively included in NEIVA largely to capture the extensive speciation of gas- and particle-phase organic carbon (i.e., NMOC_g and NMOC_p) that has been achieved in laboratory studies. Laboratory studies also provide additional measurements

for fuel and fire types that have a limited number of field-based EF measurements, and thus if representative, decrease the uncertainty associated with those EFs. While the designation of representative is subjective, studies were prioritized here that emphasized careful handling of relevant fuels (e.g., using fresh fuels from specific locations) and combustion in configurations that mimic natural conditions to the extent possible. Nonetheless, even in these representative

laboratory studies, MCE values were typically higher than observed in the field. Therefore, the laboratory-based EFs for all fuels (with the exception of peat) were adjusted to account for the generally lower combustion efficiencies under field conditions. Briefly, to calculate the adjusted laboratory-based EFs, the laboratory-based ERs to CO were multiplied by the field-average $EF_{CO}$ for smoldering compounds; an analogous calculation was done for flaming compounds using

$EF_{CO2}$. The adjustments are described in further detail in S4. Results of the adjustment are shown here and in S4.

Figure 2 illustrates the magnitude of the adjustment to laboratory-based EFs for smoldering dominant compounds. For each fuel or fire type, the average field-based EF for CO is shown in

dark grey and the laboratory-based EF for CO in light grey. The laboratory-based CO values are lower for most fuel or fire types, with the exception of boreal forest, charcoal burning, and crop



residue. For boreal forest, the relatively high laboratory-based CO value is largely driven by EFs measured in boreal peat studies and reported by Yokelson et al. (1997). For crop residue, the relatively high value is driven by laboratory-based pile burns of rice straw reported by Christian et al. (2003). For charcoal burning, there are a greater number of field studies (n = 5) than laboratory studies (n = 2) and the variability is larger for the field studies, with lower end CO values of 122 g/kg. The sum of the adjusted EFs for the smoldering dominant compounds thus increases for most fuel or fire types, consistent with the lower $EF_{CO}$ values measured under more flaming conditions in laboratory studies. For two fire types, boreal and temperate forest, the sum of the adjusted EFs does not decrease and increase (respectively) as expected. The reason for this is twofold: the number of compounds measured in laboratory studies is significantly larger than the number measured in the field and, in the case of temperate forest, the natural variability (driven by fuel and fire characteristics) is larger than the small difference between the average field and laboratory $EF_{CO}$. Figure S2 is the analogous figure for the flaming dominant compounds (NO, $NO_2$, $NO_x$ as NO, $N_2O$, HONO, $SO_2$, HCl, gaseous Hg).

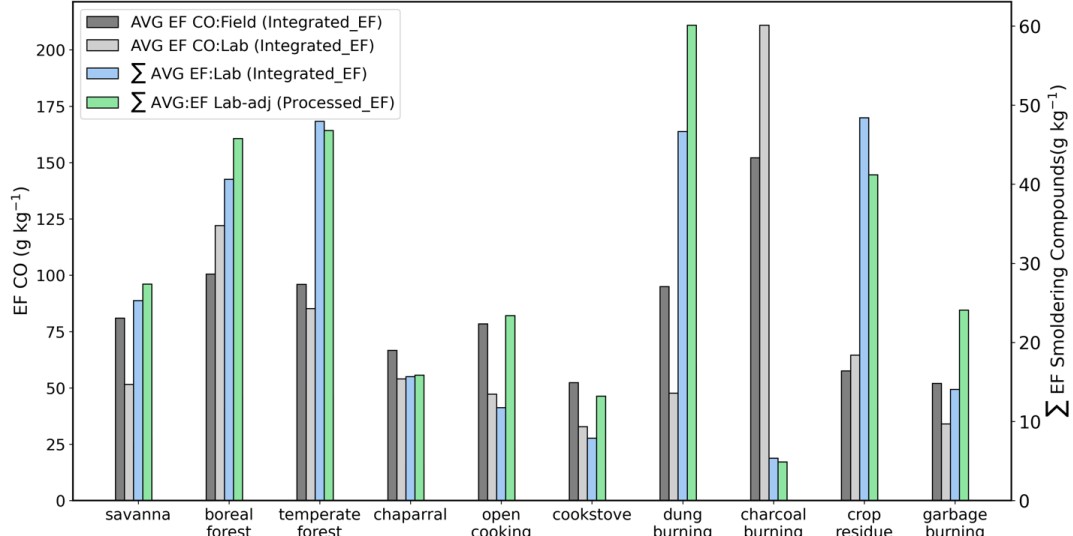

**Figure 2: Averaged EF values for CO (field, dark grey; lab, light grey) and the sum of smoldering dominant compounds (excluding CO and CH₄) pre-(blue) and post-(green) adjustment to account for differences in combustion conditions between laboratory and field studies. Integrated_EF indicates data from the integrated EF dataset and Processed_EF indicates data from the processed EF dataset.**

In the processed EF dataset the adjusted laboratory-based EFs replace the unadjusted laboratory-based EFs from the integrated dataset and are used in the calculation of the recommended EFs. To more closely evaluate this adjustment on an individual compound level, Figure 3 shows the distribution of field and adjusted laboratory EFs (box and whiskers) for the 25 most abundant NMOC_g in the temperate forest fire type. The mean value is equivalent to the recommended EF and is shown by the red line. Also shown are the average EF based on the unadjusted laboratory data only ('Average EF (lab)') and the field data only ('Average EF (field)'), as well as the EFs reported by Permar et al. (2021) for WE-CAN and Gkatzelis et al. (2023) for FIREX-AQ. A corresponding figure for the 25 compounds with the highest number of observations ("n") in the NEIVA integrated EF database, that are not shown in Fig. 3, is included in the SI (Figure S3), and equivalent figures for crop residue are also included in the SI (Figures S4, S5). While the



unadjusted laboratory EF averages are outside the upper (75%) and lower (25%) quartiles for five of the 25 compounds shown in Fig. 3 (and 11 of the 25 in Fig. S5), the mean EF values (which include adjusted laboratory EF) do not deviate significantly from the field-based averages. Agreement with the values reported by Permar et al. (2021) and Gkatzelis et al. (2023) is compound dependent, but generally those values are within the upper and lower quartiles of the NEIVA processed dataset. This analysis suggests that the inclusion of the adjusted laboratory data does not introduce unrepresentative values that are outside of the expected variability and/or uncertainty observed in the field data, and serves to increase the number of observations and compounds represented in the database.

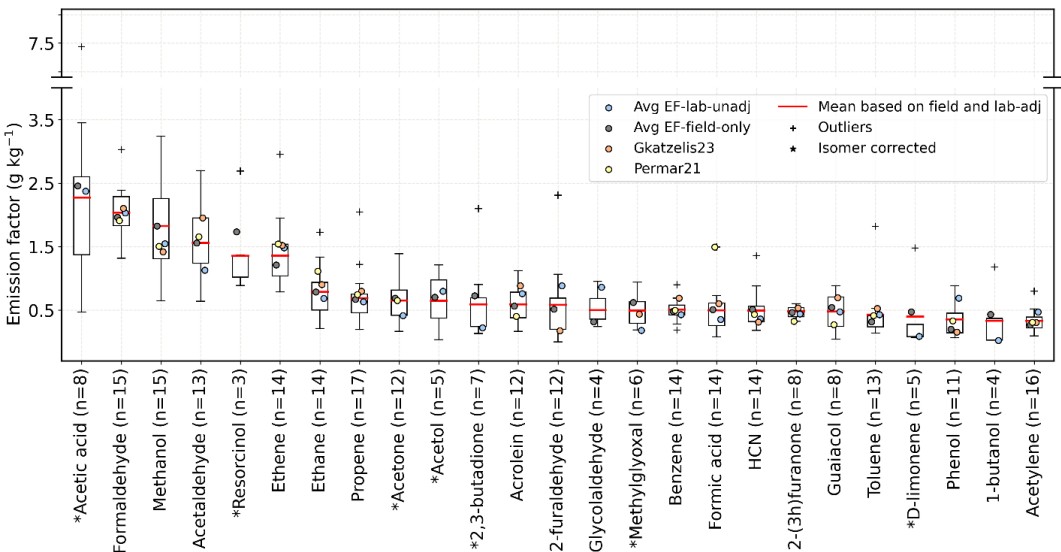

**Figure 3: The 25 most abundant NMOC_g EFs for temperate forest. The box and whiskers represent the values in the processed EF dataset and thus include the field EFs and the adjusted laboratory EFs. The outliers (> 1.5 × above/below the interquartile range) in the processed EF dataset are indicated by the plus symbols. The red line indicates the mean value and is equivalent to the recommended EF. The number of observations is listed in parenthesis ("n"). Compounds marked with an asterisk before the name have had an additional correction, application of isomeric distributions described below.**

## 4.2 Assignment of Isomer Contributions to Exact Masses

In some cases, isomers that are not resolved using one analytical technique can be resolved using another analytical technique. Because the individual compounds in these unresolved mixtures may have very different chemical and physical properties, it is preferable to resolve the mixtures when possible. In addition, resolving mixtures leads to an increase in the number of observations for associated individual compounds. Therefore, prior to their inclusion in the recommended EF database, fractional distributions were assigned to mixtures as described in S4.

The summed EFs for groups of NMOC_g in the NEIVA integrated dataset that were assigned fractional distributions are listed before and after processing in Table 4 for each fuel or fire type. Also included are the number of unique chemical formulas for which isomer contributions were assigned. The summed EFs for these NMOC_g decreases with the application of the fractional distribution, largely due to double counting prior to assigning isomer contributions to groups of



NMOC_g. There were no group assignments in open cooking or charcoal making so no isomer
565    contribution assignments were made.

**Table 4:** The summed EFs for the subset of NMOC_g to which isomeric contributions have been assigned, pre- and post- assignment of fractional contributions, shown for each fuel or fire type. Also shown are the number of unique chemical formulas for which fractional distributions were assigned.

| Fuel or Fire Type | Summed Isomeric NMOC_g EFs Pre-Fractional Contribution | Summed Isomeric NMOC_g EFs Post-Fractional Contribution | Number of Unique Chemical Formulas |
|---|---|---|---|
| Savanna | 12.23 | 7.02 | 11 |
| Boreal forest | 8.33 | 4.16 | 38 |
| Tropical forest | 1.47 | 0.74 | 2 |
| Temperate forest | 28.27 | 14.71 | 80 |
| Peat | 37.44 | 19.67 | 76 |
| Chaparral | 11.74 | 5.93 | 36 |
| **Domestic Biomass Burning** | | | |
| Open cooking | 0 | 0 | 0 |
| Cookstove | 0.47 | 0.20 | 4 |
| Dung burning | 22.56 | 12.22 | 15 |
| Charcoal making | 0 | 0 | 0 |
| Charcoal burning | 0.90 | 0.42 | 1 |
| Pasture maintenance | 0.17 | 0.09 | 1 |
| Crop residue | 28.05 | 13.93 | 85 |
| Garbage burning | 2.33 | 1.21 | 6 |

The laboratory-based EFs in the processed EF dataset were adjusted for MCE and, where
applicable, assigned isomeric contributions. Figures 4 and 5 compare the NEIVA temperate forest
570    EFs from the recommended EF database (includes laboratory-adjusted EFs) with EFs reported by
Permar et al. (2021) for WE-CAN and Gkatzelis et al. (2023) for FIREX-AQ, respectively. For
115 of 145 overlapping gaseous compounds agreement is within a factor of two with Permar et al.
(2021) and for 84 of 95 with Gkatzelis et al. (2023). Focusing on the compounds for which NEIVA
is higher than Permar et al. (2021)and/or Gkatzelis et al. (2023) by a factor of two or more, there
were no systematic biases or unexplained discrepancies in the laboratory data relative to the field
data, supporting the inclusion of laboratory data in this EF compilation. For some compounds,
higher EFs measured in laboratory studies, and in Gkatzelis et al. (2023) relative to Permar et al.
(2021), can be explained by photochemical losses as a function of aging. In Figs. 4 and 5, marker
colors are representative of $k_{OH}$ values for the NMOC_g, with red values indicating higher OH
reactivity and blue values indicating lower OH reactivity. The loss of the more reactive compounds
measured during WE-CAN relative to laboratory studies likely partially explains the higher EFs
in NEIVA, and to a lesser extent the compounds measured during FIREX-AQ. Similar
observations were made by Gkatzelis et al. (2023), that ERs for some highly reactive compounds
in WE-CAN were lower than laboratory measurements and in FIREX-AQ higher than laboratory
measurements, highlighting variability in oxidation and emissions in both laboratory and field
studies. When multiple data points were available for comparison, high EF values were also
reported for field studies (and low EF values for laboratory studies) representing diversity in fuels





burned and fires sampled. For some field studies, the higher EFs reflect greater sampling of smoldering fires (e.g., as reported by Yokelson et al. (2013)) and pile burns (e.g., as reported by Travis et al., 2023)). No laboratory data were omitted as a result of these comparisons.

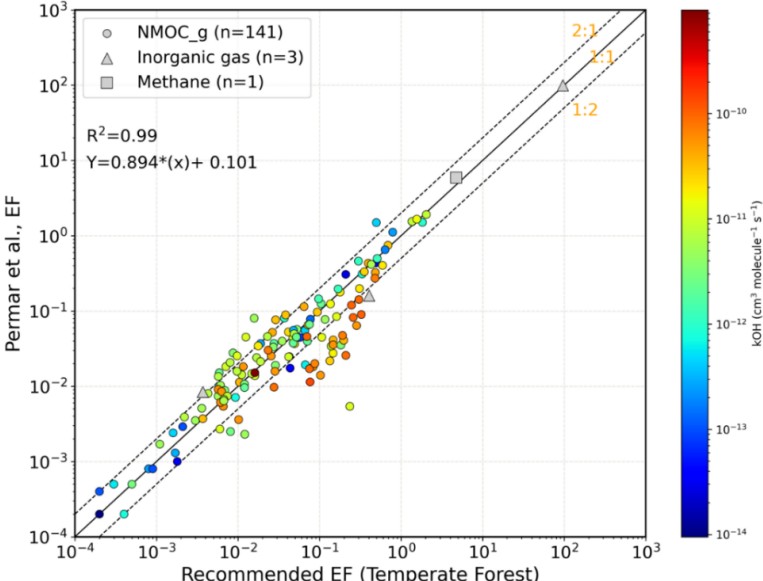

**Figure 4: NEIVA temperate forest EFs (gaseous non-methane organic compounds, inorganic gases, methane) vs. EF data reported by Permar et al. (2021) from the WE-CAN field study. The equation is for the linear fit (not shown).**

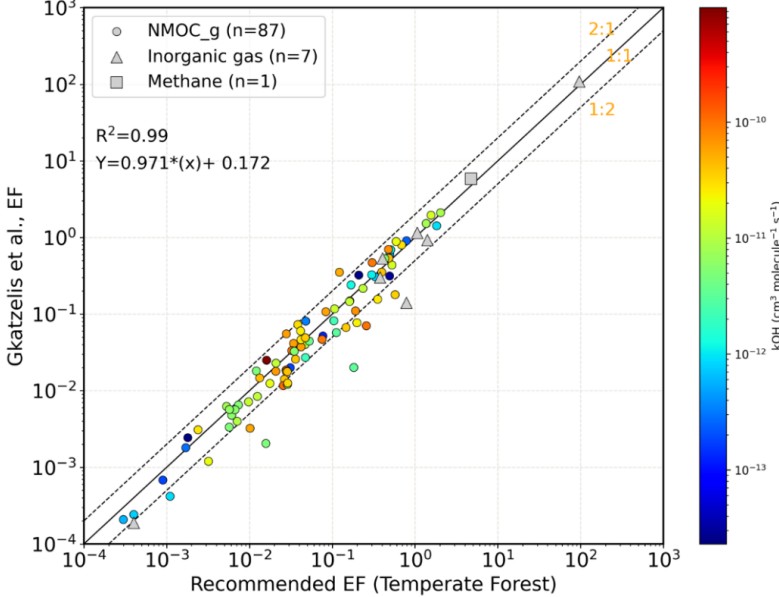

**Figure 5: NEIVA temperate forest EFs (gaseous non-methane organic compounds, inorganic gases, methane) vs. EF reported by Gkatzelis et al. (2023) from the FIREX-AQ field study. The equation is for the linear fit (not shown).**





### 4.3    Comparisons of Recommended EFs with EF Compilations of Akagi et al. 2011 and Andreae 2019

In 2019, Andreae (2019) published an update of the 2001 Andreae and Merlet (2001) EF compilation. Field data from over 370 publications were evaluated and the number of species included was increased from 93 to 121. Andreae (2019) compared EFs for a subset of compounds and constituents with Akagi et al. (2011). That comparison is expanded here, with an added emphasis on NMOC_g. Figure 6 is similar to Figure 2 of Andreae (2019) and shows a comparison of NEIVA-based recommended EFs for selected inorganic gases and particulate constituents with
Akagi et al. (2011) (green markers) and with Andreae (2019) (orange markers) for three fire types (represented by the different marker symbols). There appear to be no systematic biases with regard to specific EFs and specific fuel types. For many of the comparisons shown, the agreement is within a factor of two (indicated by the dashed lines). The methane EF for crop residue in the NEIVA recommended EF dataset is lower than both Andreae (2019) and Akagi et al. (2011) likely
due to the inclusion of more data from loose burning in the field. In addition, the OC EFs are higher than Andreae (2019) for crop residue, which is likely due to inclusion of the Travis et al. (2023) data, in which the burns occurred under relatively wet conditions, promoting more smoldering combustion. The BC EFs in the NEIVA recommended EF dataset are lower than Andreae (2019) for temperate forest and significantly so for peat. The significantly lower BC EFs for peat in the
NEIVA recommended EF dataset are largely due to exclusion of thermal EC data, which can result in artificially high EC/BC EFs due to charring of OC. Figures S6-S13 show additional comparisons between NEIVA and Andreae EF datasets for the most abundant compounds in temperate forest, peat, and crop residue fire types.

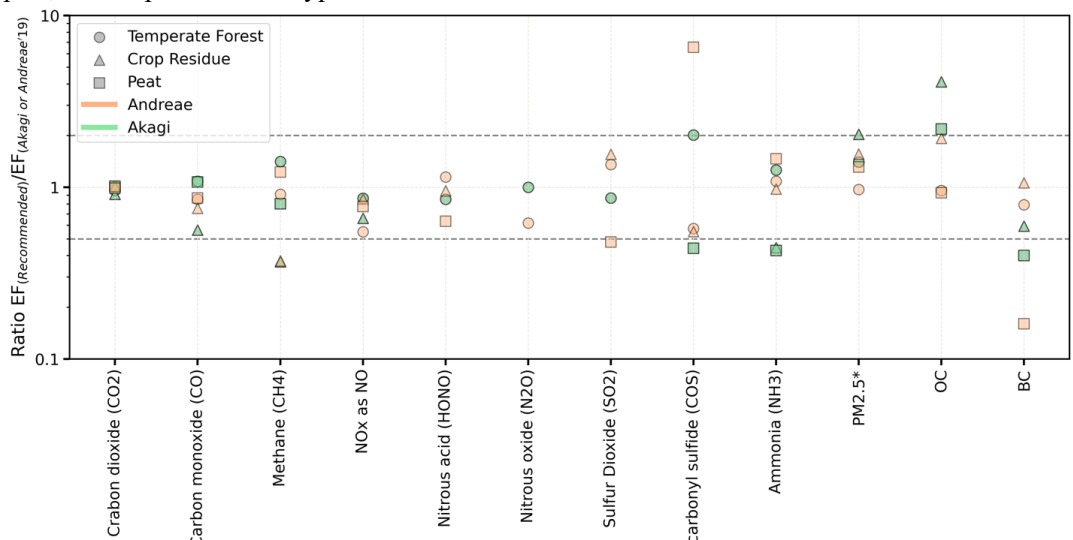

**Figure 6: Ratio of recommended EFs based on NEIVA to EFs based on Akagi et al. (2011) (orange) and Andreae (2019) (green) to for selected gases and particulate constituents in temperate forest, crop residue, and peat fire types. Agreement within a factor of two is shown by the dashed lines; PM$_{2.5}$* includes PM$_{1-5}$.**



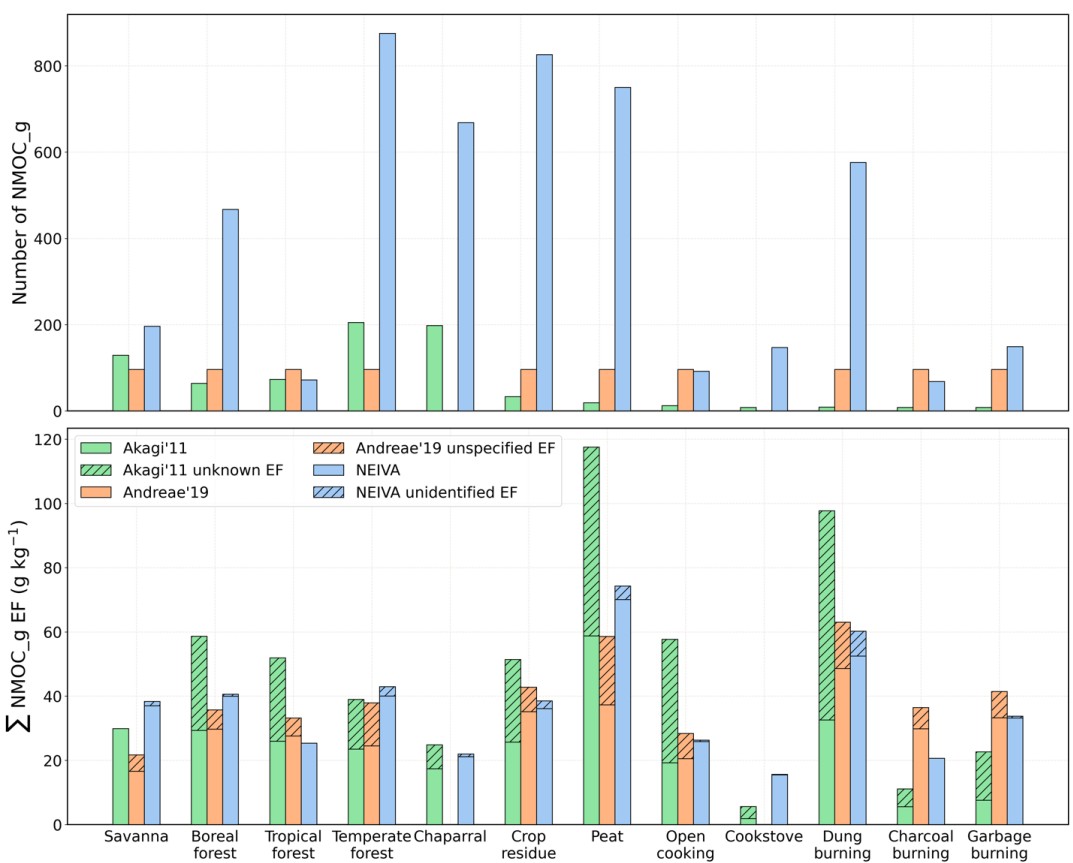

**Figure 7: Number of compounds represented in NMOC_g (top panel) and total NMOC_g EF (bottom panel) in Akagi et al. (2011), Andreae (2019), and NEIVA. Andreae (2019) reported the total NMOC_g EF from Akagi et al. (2011); here, the total NMOC_g EF based on Andreae (2019) is the sum of individually reported NMOCs plus reported non-specified "VOC" (the latter shown by hashes).**

The inclusion of laboratory data in NEIVA leads to an unprecedented increase in the number of individual NMOC_g represented for globally-relevant fuels and fire types. In Figure 7, the number of NMOC_g (top panel) and total NMOC_g EF (bottom panel) are compared with Akagi et al. (2011) and with Andreae (2019) across the 12 fuel and fire types updated in NEIVA (pasture maintenance and charcoal-making were not updated). Andreae (2019) does not include data for two of these fire types, chaparral (shrubland) and cookstoves. While the previously published compilations include approximately 100-200 NMOC_g for most fire types (excluding cookstoves), NEIVA includes more compounds in nine of the fourteen fire types, with > 400 NMOC_g for six of the fire types. Further, except for tropical forest, the increase in the number of NMOC_g represented nearly eliminates the unknown NMOC_g EF approximated by Akagi et al. (2011) (the total of which was also reported by Andreae (2019)). The differences between the total NMOC_g EF based on Akagi et al. (2011) and based on NEIVA largely arise from the extent to which this unknown fraction was under- or over-estimated (which has not been investigated for tropical forest since Akagi et al. (2011)). For a few less-sampled fire types, Andreae (2019) has a slightly higher total EF NMOC_g than NEIVA due to inclusion of summed non-specified VOCs.





In NEIVA there is still some fraction of NMOC_g, ≤ 5% for most fire types, for which the molecular formula is known but compound class cannot be assigned ("unidentified").

In Fig. 7, it can be seen that for some fire types (e.g., boreal forest, crop residue, dung burning) although the number of NMOC_g EF represented in NEIVA increases by a factor of four or more, the NMOC_g EF is less than the Akagi et al. (2011) total including estimated unknowns. In Figures 8-10, the total number of compounds that are required to represent 90% of the NMOC_g EF in NEIVA is shown for boreal forest, crop residue, and dung burning, respectively. Analogous figures

for other fuel and fire types are in the SI (S14-S17). The threshold of 90% was chosen arbitrarily. The figures illustrate that inclusion of ~100 compounds represents the majority of the total NMOC_g EF, and thus the NMOC_g EFs in Fig. 7b vary less than the number of compounds in Fig.7a. Although a large number of compounds have small EFs, collectively they represent a non-negligible fraction of the total NMOC_g. Further, some representation of their chemical and

physical properties will be required for accurate predictions of smoke composition and concentration and of the effects of smoke on atmospheric composition, air quality, and climate.

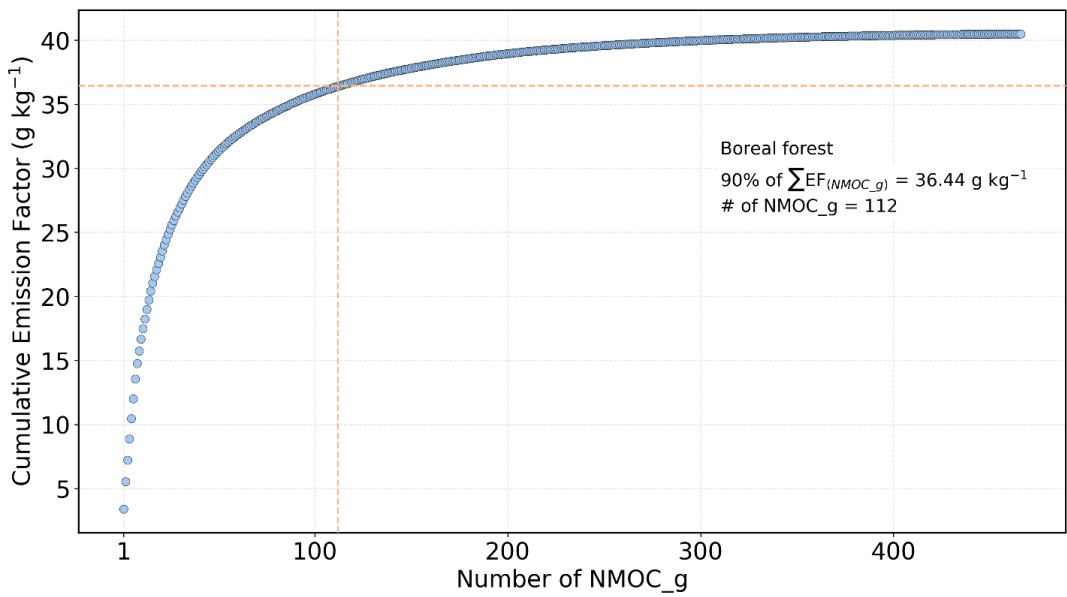

**Figure 8: Number of compounds needed to represent 90% of the total boreal forest NMOC_g EF.**



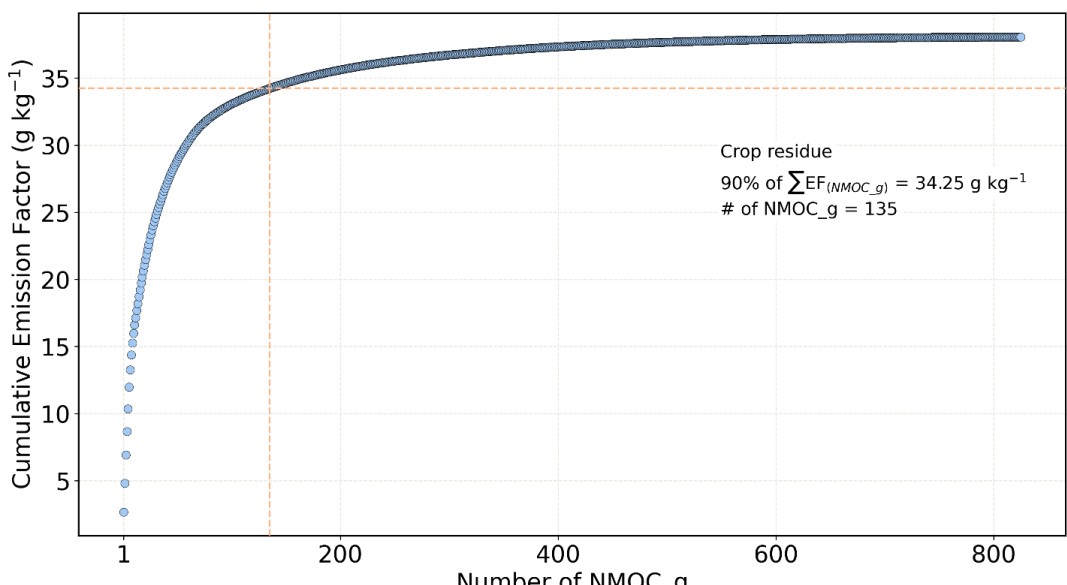


**Figure 9: Number of compounds needed to represent 90% of the total crop residue NMOC_g EF.**

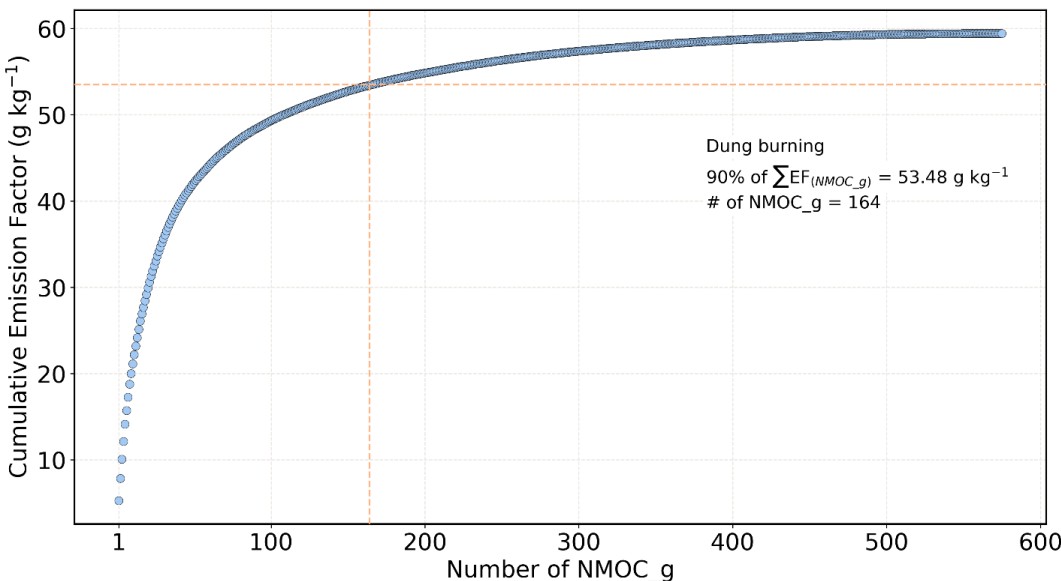

**Figure 10: Number of compounds needed to represent 90% of the total dung burning NMOC_g EF.**




A more detailed comparison between Akagi et al. (2011) and NEIVA is shown in Figure 11. The EF is summed by individual compounds that are matched between the two datasets and individual compounds that are unmatched between the two datasets (i.e., appear in the NEIVA database but not in Akagi et al. (2011)). Also shown is a total EF representing unknown compounds in Akagi et al. (2011) and unidentified compounds in NEIVA (formula known but no functional group or structural assignment). For boreal forest, the summed NMOC_g EF for matched compounds is lower in NEIVA than in Akagi due to the increased weighting of smoldering fires in Akagi et al. (2011). For temperate forest and for chaparral, the unknown EF in Akagi is similar to the unmatched EF in NEIVA, suggesting a reasonable approximation of unknowns for these fire types by Akagi et al. (2011). For crop residue, the EF for matched compounds is lower in NEIVA than in Akagi due to the reduced weighting of pile burns in NEIVA. For peat, the EF for matched compounds is lower in NEIVA than Akagi due to the inclusion of new EF data from several studies which are lower than those reported by Christian et al. (2010) and compiled in Akagi et al. (2011). There are no differences between the matched and unmatched compounds for tropical forest because no new NMOC_g data were added.



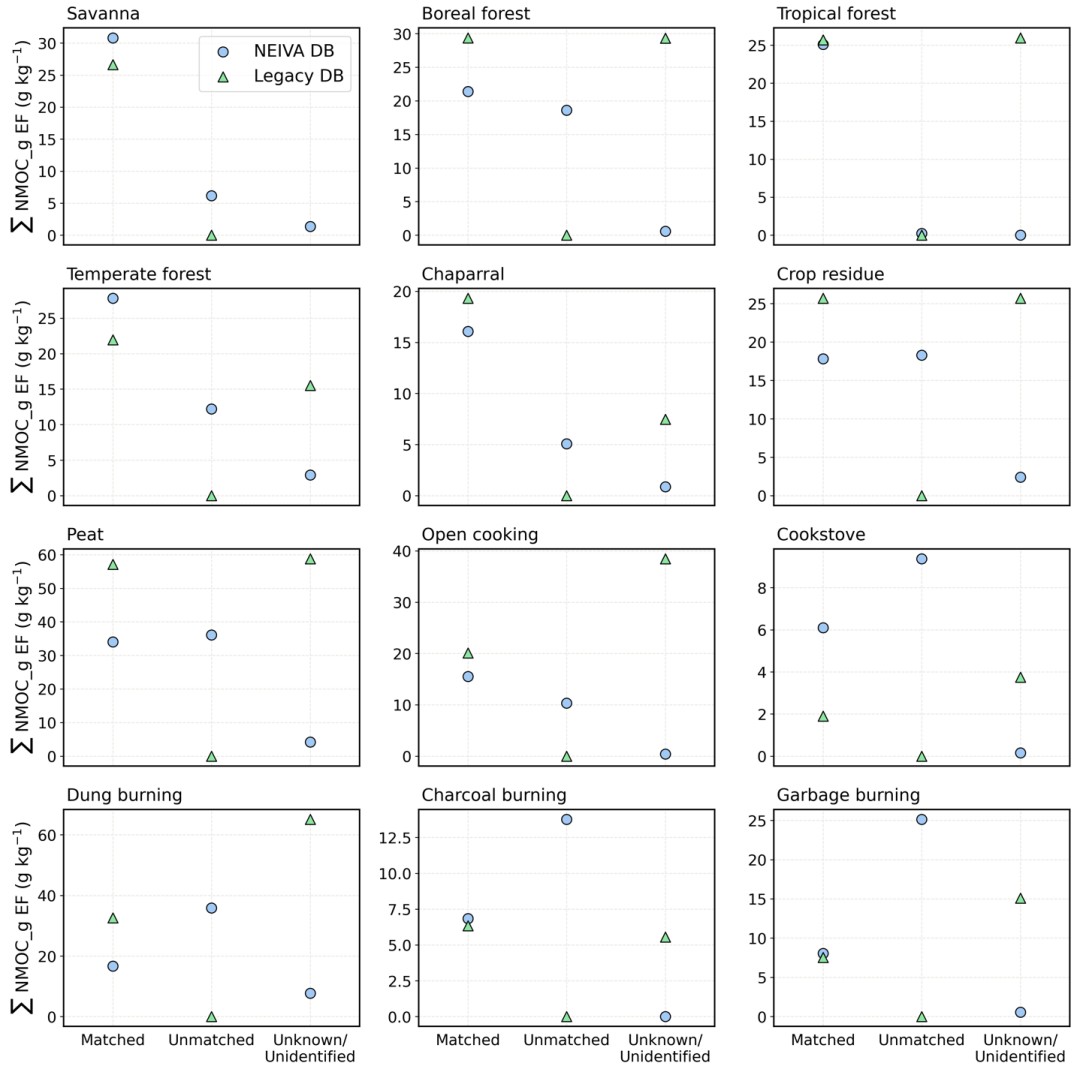

**Figure 11: Comparison of the summed NMOC_g for compounds that are matched and unmatched (i.e., in NEIVA but not in Akagi et al. (2011)) between NEIVA and Akagi et al. (2011), and the summed NMOC_g that were unknown (approximated) in Akagi et al. (2011) and are unidentified (formula but no functional group, structural assignment) in NEIVA. The middle and right-hand column of each chart compare total unknown NMOC_g mass estimated Akagi et al. (2011) to the mass of newly identified species included in this work.**





## 4.4 Implications for Atmospheric Composition and Chemistry


Representation of a greater diversity of NMOC_g has a number of potential implications for predictions of atmospheric composition, chemistry, and associated effects (e.g., Xu et al., 2021; Schwantes et al., 2022). The magnitude of the effects will depend on model complexity and resolution, and will be further investigated in forthcoming manuscripts. In lieu of a detailed modeling analysis, features of the distributions of NMOC_g are presented here that can affect predictions of atmospheric composition and chemistry. The ability to generate property distributions for individual compounds and representative model surrogates is enabled by the chemical mechanism and property dataset that are linked to the EF datasets using unique IDs.


The volatility distribution of organic compounds, represented here by decadally spaced bins of saturation vapor concentration ($C^*$), is important for predictions of SOA formation and deposition. Figures 12 and 13 show the volatility distribution of NMOC_g normalized to the total NMOG in each inventory for temperate forest and crop residue fires based on NEIVA, Andreae (2019), and the EPA SPECIATE 5.2 database (Simon et al., 2010; Bray et al., 2019; SPECIATE, 2023) for temperate forest (profile 95424) and crop residue (profile 5564). The compounds are grouped by their $C^*$ values in logarithmic bins. As demonstrated by Hatch et al. (2017), improved speciation of NMOC_g leads to inclusion of lower volatility compounds than are currently represented in emissions inventories. Relative to the NEIVA database, the distributions of compounds in Andreae (2019) and the EPA SPECIATE 5.2 database (Simon et al., 2010; Bray et al., 2019; SPECIATE, 2023) are skewed towards higher volatility bins and the intermediate volatility compounds (IVOCs, $3.5 < \log C^* < 6.5$) are underrepresented and in some cases entirely absent.




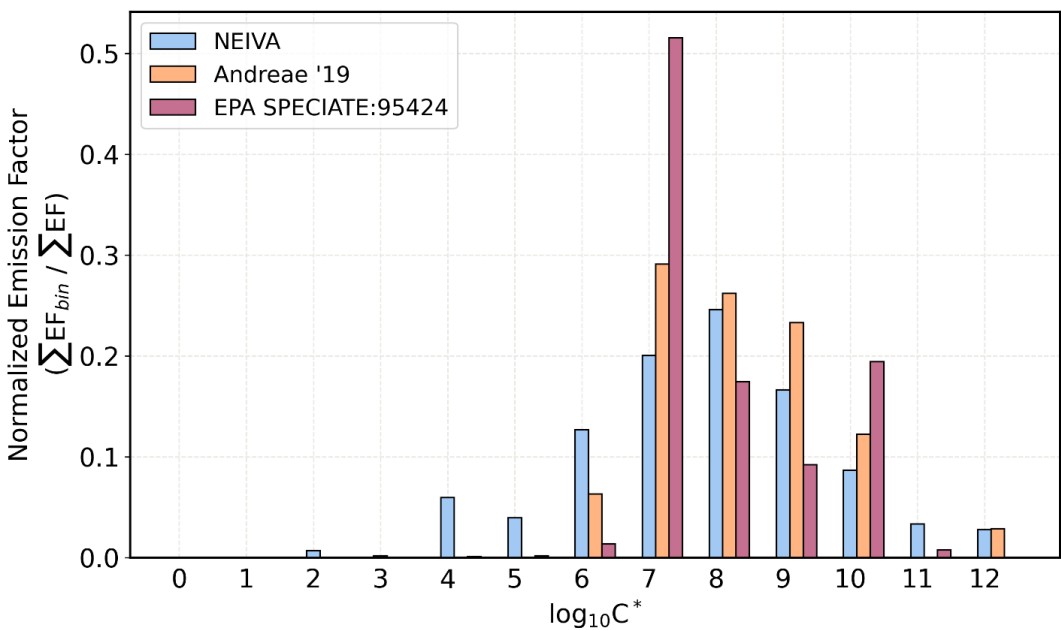

**Figure 12: Normalized volatility distribution of temperate forest NMOC_g EFs using NEIVA compared with Andreae ( 2019) and the EPA SPECIATE (Simon et al., 2010; Bray et al., 2019; SPECIATE, 2023) profile for western wildfire (#95424).**


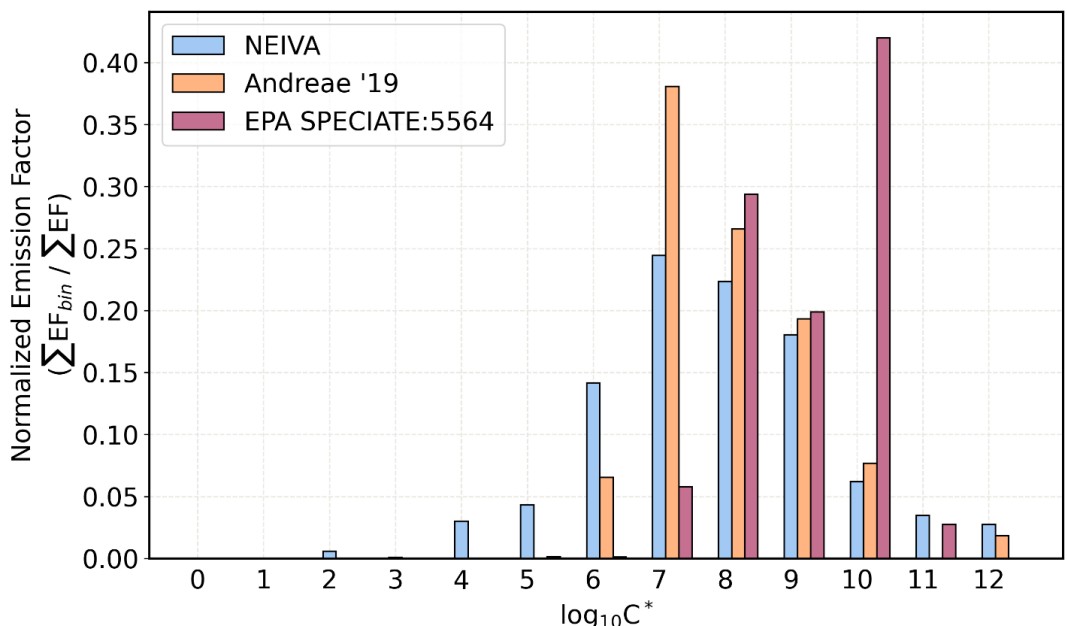

**Figure 13: Normalized volatility distribution of temperate forest NMOC_g EFs using NEIVA compared with Andreae (2019) and the EPA SPECIATE (Simon et al., 2010; Bray et al., 2019; SPECIATE, 2023) profile for crop/agriculture residue (#55644).**

For many types of modeling, while some NMOC_g are explicitly represented, most are mapped to model surrogate species that are specific to the chemical mechanism being used. In NEIVA v1.0, the NMOC_g compounds were mapped to surrogate species for the following chemical mechanisms: SAPRC-07/-07 toxics (Carter, 2010), SAPRC-18 (Carter, 2020), SAPRC-22 (Carter, 2023a); MOZART-T1(Emmons et al., 2020); and GEOS-Chem (Bey et al., 2001; Carter et al., 2022). The number of model surrogates used to represent these compounds is mechanism dependent and listed in Table S18. Figures 14 and 15 show the relative distribution, based on mole fraction, of NMOC_g mapped to SAPRC-07 model compounds for temperate forest and crop residue. The distributions shown here are independent of the number of compounds represented in each EF compilation and of the total NMOC_g EF, but are dependent on the identities of the individual compounds and their relative contributions to the total NMOC_g EF in each inventory. For compounds that are listed as "unspeciated" or "unidentified", that mass was distributed equally among the model lumped categories as is typically done in model applications, though more recently published data (e.g., Stockwell et al., 2015; Koss et al., 2018) included here suggest the unidentified species are primarily high molecular mass oxygenated species consistent with the shift in $C^*$ shown in Figs. 12 and 13.





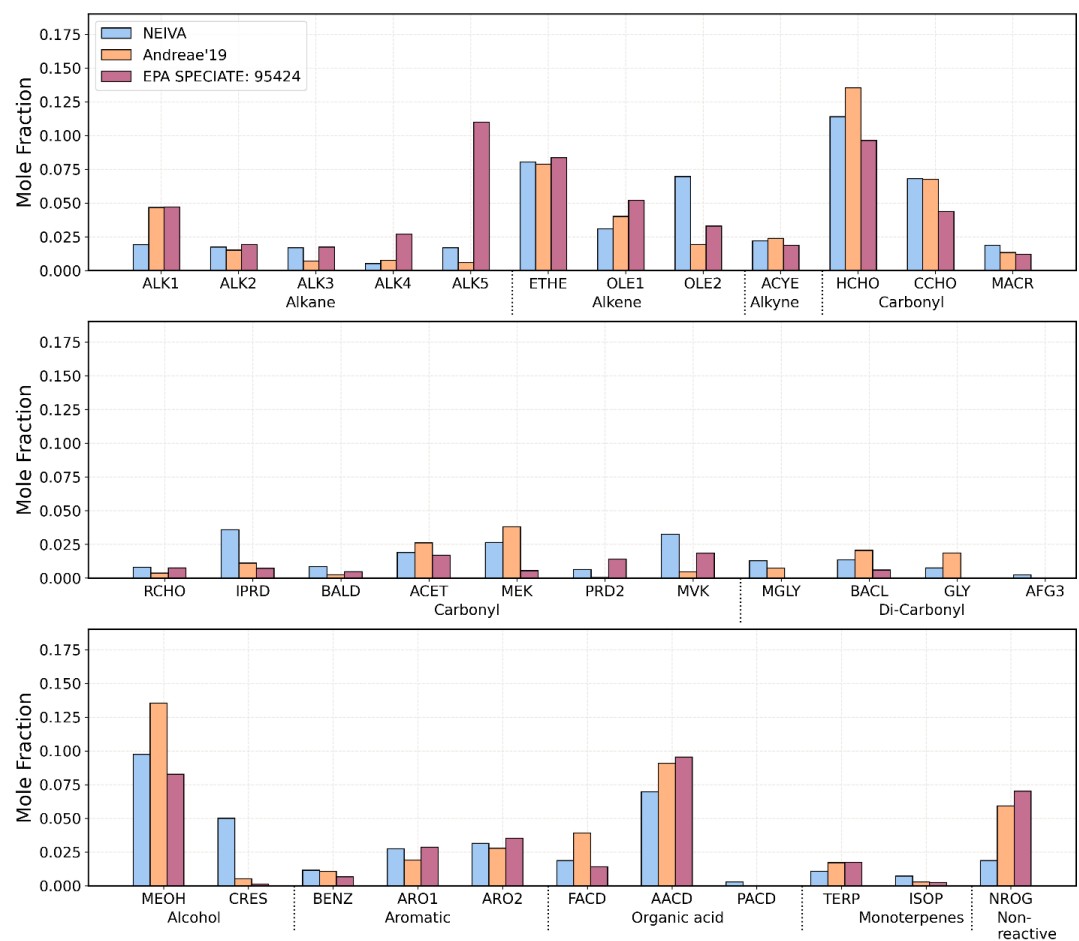

**Figure 14: NMOC_g mapped to SAPRC-07 model surrogate species based on NEIVA compared with Andreae ( 2019) and the EPA SPECIATE (Simon et al., 2010; Bray et al., 2019; SPECIATE, 2023) profile for western wildfire (95424).**



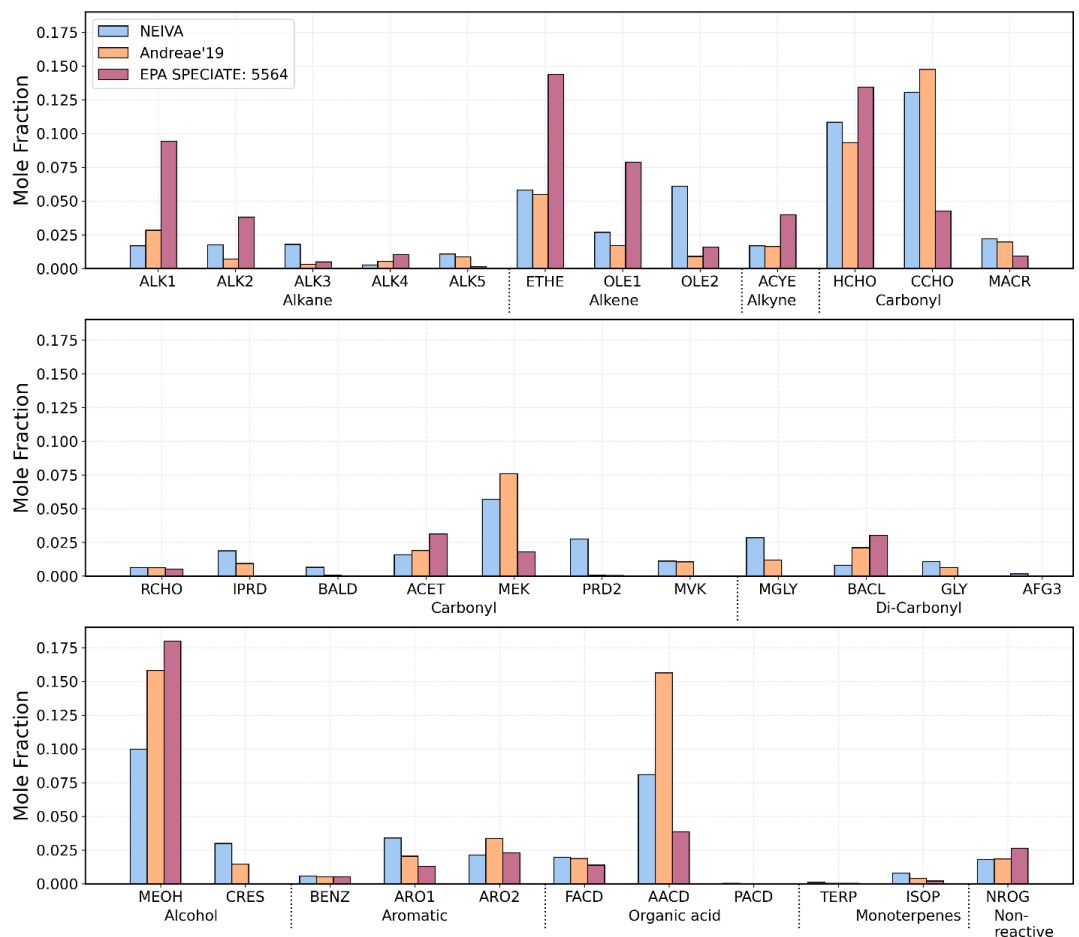

**Figure 15: NMOC_g mapped to SAPRC-07 model surrogate species based on NEIVA compared with Andreae ( 2019) and the EPA SPECIATE (Simon et al., 2010; Bray et al., 2019; SPECIATE, 2023) profile crop/agriculture residue (#55644).**

The compounds represented in NEIVA, Andreae (2019), and the EPA SPECIATE 5.2 database (Simon et al., 2010; Bray et al., 2019; SPECIATE, 2023), have distinctly different profiles when mapped to the 37 SAPRC-07 model species. Figures 16 and 17 show the calculated OH reactivity (OHR) as influenced by the different model surrogate distributions shown in Figures 14 and 15 for temperate forest and crop residue, respectively. The sizes of the charts are scaled by the total OHR ($s^{-1}$) calculated for a representative NMOC_g mixing ratio of ~90 ppb. The OH reaction rate constants were based on published literature for the respective chemical mechanisms and were not recalculated to represent the mixture of compounds mapped to each surrogate. The top 8 model species with the largest contributions to OHR are explicitly shown, and the contributions of the remaining 29 model species are summed and represented as "others". The OHR calculated using the NEIVA-based distribution of model compounds is ~50-60% and ~60-90% higher than the OHR calculated using the Andreae (2019) and the EPA SPECIATE (Simon et al., 2010; Bray et al., 2019; SPECIATE, 2023) distributions for temperate forest and crop residue, respectively. This is largely driven by the greater mole fractions of model species OLE2 (more reactive alkenes, $k_{OH}$





$> 4.8 \times 10^{-11}\,cm^3\,molec^{-1}\,s^{-1}$) and IPRD (unsaturated aldehydes) in both fire types, and additionally CRES (oxygenated aromatic hydrocarbons including phenols and cresols, but not furan or furan derivatives) in temperate forest.

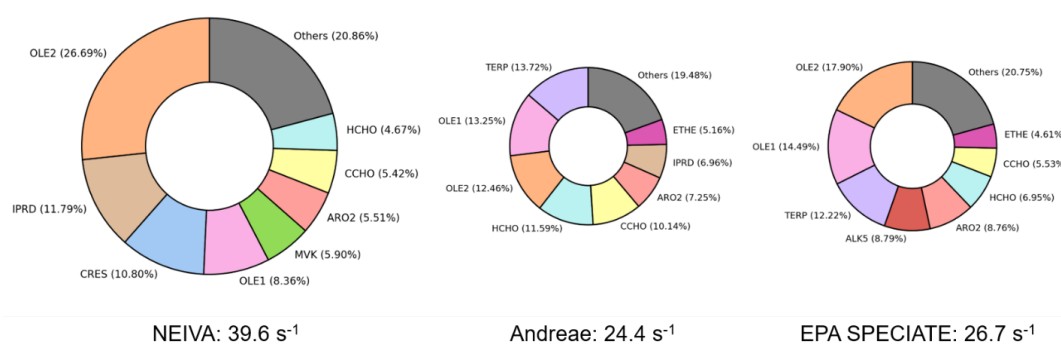


**Figure 16: OH reactivity calculated using the surrogate species distributions in Fig. 14; chart size is scaled to the OH reactivity value.**

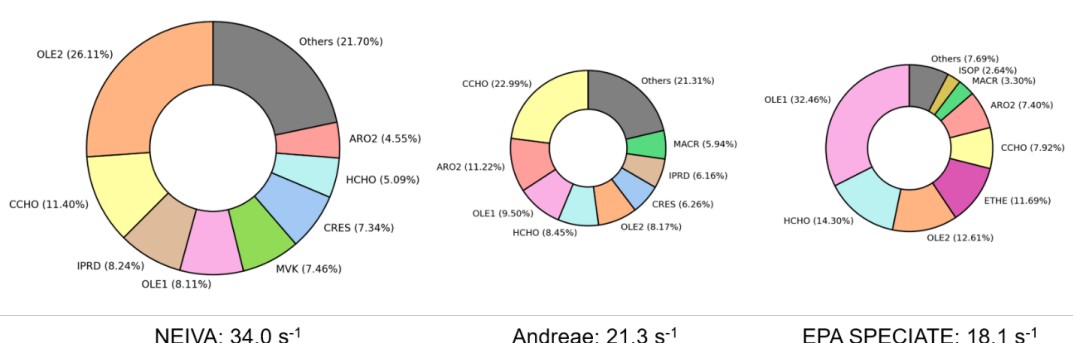


**Figure 17: OH reactivity calculated using the surrogate species distributions in Fig. 15; chart size is scaled to the OH reactivity value.**

## 5. Database Products

The NEIVA GitHub repository includes all of the database files in '.sql' and '.csv' format, and associated Python scripts (executable using the Python package `neivapy`) that were used to create the datasets, which also can be used to create new datasets upon the addition of new data, and to query the datasets. Jupyter notebooks are additionally shared in the NEIVA GitHub repository that demonstrate the features of the database, including adding new data and generating new datasets (`add_new_data.ipynb`) and example functions for querying the
data(`NEIVA_query_mysql.ipynb`,   `NEIVA_py_functions.ipynb`).   These notebooks allow users to setup the NEIVA database in a Google Colab environment, execute



MySQL syntax, apply the `neivapy` functions, and download data. A full list of functions is provided in S9 (see Table S24). Some example functions are shown below.

## 5.1 Display information

The functions highlighted in this section are used to access and display information and labels.

`table_info(database, fire_type)`. This function returns a list of table names along with associated information such as measurement type, publication DOI, pollutant category for a specified database name (legacy database (ldb), raw database (rdb), and primary database (pdb), 785 in acronym format (ldb, rdb, pdb) and fire type.

**Table 5:** The output of the `table_info()` function when using the parameters- rdb, garbage burning.

| table_name | measurement_type | fire_type | pollutant_category | study | source | doi |
|---|---|---|---|---|---|---|
| rdb_gb_yokelson13 | lab | garbage burning | inorganic gas, methane, NMOC_g, PM size | yokelson13 | Table S1 | doi.org/10.5194/acp-13-89-2013 |
| rdb_goetz18 | field | dung burning, cookstove, crop residue, garbage burning, charcoal burning | PM size, PM organic, PM elemental, PM ion | goetz18 | Supplement section 3 and 4 | doi.org/10.5194/acp-18-14653-2018 |
| rdb_jayarathne18 | field | garbage burning, cookstove, dung burning, crop residue, open cooking peat | PM size, PM organic, PM elemental, PM ion, PM metal, NMOC_p | jayarathne18 | Table 2, 3.2 Emission of OC, EC and WSOC | doi.org/10.5194/acp-18-2585-2018 |
| rdb_stockwell15 | lab | crop residue, boreal forest, chaparral, cookstove, open cooking, temperate forest, peat, garbage burning | inorganic gas, methane, NMOC_g | stockwell15 | Table S2 | doi.org/10.5194/acp-15-845-2015 |
| rdb_stockwell16 | lab, field | dung burning, cookstove, open cooking, charcoal burning, crop residue, garbage burning, peat | inorganic gas, methane, NMOC_g, PM elemental, PM optical property | stockwell16 | Table S8, Table S7, Table S9, Table 6 | doi.org/10.5194/acp-16-11043-2016 |

`summary_table(fire_type, measurement_type)`. This function returns a list of emission factor column names in the integrated EF table along with information such as MCE, 790 measurement type (lab or field study), fuel type, and additional information for specific fire types if available (e.g., cookstove name).

**Table 6:** The output of the `summary_table()` function when using the parameters- peat, field.

| efcol | measurement_type | MCE | fuel_type | study |
|---|---|---|---|---|
| EF_peat_jayarathne18 | field | 0.78 | indonesian peat | jayarathne18 |
| EF_tropical_peat_roulston18 | field | 0.83 | indonesian peat | roulston18 |
| EF_tropical_peat_smith17 | field | 0.80 | indonesian peat | smith17 |
| EF_peat_stockwell16 | field | 0.77 | indonesian peat | stockwell16 |
| EF_peat_north_carolina_pokhrel16 | field | 0.72 | north carolina peat | pokhrel16 |
| EF_peat_canada_pokhrel16 | field | 0.80 | canada peat | pokhrel16 |
| EF_peat_indonesia_pokhrel16 | field | 0.81 | indonesian peat | pokhrel16 |





### 5.2 Query emission factor data


The functions highlighted here are used for querying EF data.

`select_pm_data(fire_type, table_name)`. This function returns the EFs in all PM subcategories (e.g., PM size, PM organic, PM elemental, PM ion, PM metal, NMOC_p and PM optical property) for the specified fire type. In the example below, tables are separated for easier viewing and PM metal and NMOC_p tables are in the SI (S9) due to their length.


**Table 7:** The output of `select_pm_data()` function when using the parameters- peat, integrated EF. The pollutant category-PM size is presented.

| EF columns | PM2.5 |
|---|---|
| EF_peat_jayarathne18 | 1.73E+01 |
| EF_tropical_peat_roulston18 | 2.77E+01 |
| EF_russia_watson19 | 4.26E+01 |
| EF_siberia_watson19 | 3.39E+01 |
| EF_northern_alaska_watson19 | 2.40E+01 |
| EF_evergladesNP_florida_watson19 | 2.36E+01 |
| EF_malaysia_watson19 | 2.24E+01 |

**Table 8:** The output of `select_pm_data()` function when using the parameters- peat, integrated EF. The pollutant category- PM organic is presented.

| EF columns | OC | water-soluble OC fraction | water-insoluble OC fraction |
|---|---|---|---|
| EF_akagi11_indonesian_peat_christian03 | 6.02E+00 | | |
| EF_peat_jayarathne18 | 1.24E+01 | 1.98E+00 | 1.04E+01 |
| EF_peat_stockwell16 | | | |
| EF_russia_watson19 | 2.51E+01 | 1.55E+01 | |
| EF_siberia_watson19 | 2.60E+01 | 8.65E+00 | |
| EF_northern_alaska_watson19 | 1.74E+01 | 6.69E+00 | |
| EF_evergladesNP_florida_watson19 | 1.90E+01 | 7.76E+00 | |
| EF_malaysia_watson19 | 1.80E+01 | 3.60E+00 | |

**Table 9:** The output of `select_pm_data()` function when using the parameters- peat, integrated EF. The pollutant category- PM elemental is presented.

| EF columns | BC | EC |
|---|---|---|
| EF_akagi11_indonesian_peat_christian03 | 4.00E-02 | |
| EF_peat_jayarathne18 | | 2.40E-01 |
| EF_peat_stockwell16 | 1.00E-02 | |
| EF_russia_watson19 | | 7.70E-01 |
| EF_siberia_watson19 | | 6.90E-01 |
| EF_northern_alaska_watson19 | | 6.00E-01 |
| EF_evergladesNP_florida_watson19 | | 7.80E-01 |
| EF_malaysia_watson19 | | 2.80E-01 |




**Table 10:** The output of `select_pm_data()` function when using the parameters- peat, integrated EF. The pollutant category- PM ion is presented.

| mm | formula | compound | EF_northern_alaska_watson19 | EF_evergladesNP_florida_watson19 | EF_malaysia_watson19 |
|---|---|---|---|---|---|
| 88.02 | Cl- | chloride | 5.77E-02 | 5.64E-02 | 3.02E-02 |
| 35.45 | NO3- | nitrate | 4.60E-02 | 4.38E-02 | 2.36E-02 |
| 62.01 | O4P-3 | phosphate | | | |
| 94.97 | O4S-2 | sulfate | 8.24E-02 | 1.81E-01 | 3.36E-02 |
| 96.07 | Na | sodium | | | |
| 22.99 | H4N+ | ammonium | 1.58E-02 | 6.00E-04 | 5.00E-04 |
| 18.04 | K | potassium | 9.90E-03 | 6.20E-03 | 8.30E-03 |
| 39.10 | Mg | magnesium | | | |
| 24.31 | Ca | calcium | 6.40E-03 | | 6.00E-04 |
| 40.08 | Na+ | sodium ion | 7.30E-03 | 7.90E-03 | 3.80E-03 |
| 22.99 | K+ | potassium ion | 7.40E-03 | 1.61E-01 | 8.60E-03 |
| 39.10 | Mg+2 | magnesium ion | | | |
| 24.31 | Ca+2 | calcium ion | | | |
| 40.08 | Cl2 | chlorine | 3.32E-02 | 5.56E-02 | 1.80E-02 |


**Table 11:** The output of `select_pm_data()` function when using the parameters- peat, integrated EF. The pollutant category- PM ion is presented. (continued)

| mm | formula | compound | EF_peat_jayarathne18 | EF_russia_watson19 | EF_siberia_watson19 |
|---|---|---|---|---|---|
| 88.02 | Cl- | chloride | 7.27E-02 | 8.72E-02 | 4.62E-02 |
| 35.45 | NO3- | nitrate | 2.80E-03 | 7.58E-02 | 4.68E-02 |
| 62.01 | O4P-3 | phosphate | | | |
| 94.97 | O4S-2 | sulfate | 2.44E-02 | 9.50E-02 | 9.52E-02 |
| 96.07 | Na | sodium | | | |
| 22.99 | H4N+ | ammonium | 8.82E-02 | 5.02E-02 | 7.50E-03 |
| 18.04 | K | potassium | | 1.47E-02 | 4.58E-02 |
| 39.10 | Mg | magnesium | | | |
| 24.31 | Ca | calcium | | 1.07E-02 | |
| 40.08 | Na+ | sodium ion | 9.00E-04 | 7.10E-03 | 1.41E-02 |
| 22.99 | K+ | potassium ion | 4.50E-03 | 2.98E-02 | 7.30E-03 |
| 39.10 | Mg+2 | magnesium ion | | | |
| 24.31 | Ca+2 | calcium ion | | | |
| 40.08 | Cl2 | chlorine | | 6.30E-02 | 3.26E-02 |

**Table 12:** The output of `select_pm_data()` function when using the parameters- peat, integrated EF. The pollutant category- PM optical property is presented.

| compound | EF_peat_stockwell16 | EF_peat_north_carolina_pokhrel16 | EF_peat_canada_pokhrel16 | EF_peat_indonesia_pokhrel16 | EF_peat_kalimantan_mixed_sites_selimovic18 |
|---|---|---|---|---|---|
| EF Babs 870 (m2 kg 1) | 2.61E-02 | | | | 1.23E-02 |
| EF Bscat 870 (m 2 kg 1) | 1.83E+01 | | | | 3.14E+00 |
| EF Babs 405 (m2 kg1) | 1.35E+00 | | | | |
| EF Bscat 405 (m2 kg1) | 5.06E+01 | | | | |
| EF Babs 405 just BrC (m2 kg1) | 1.30E+00 | | | | |
| EF Babs 405 just BC (m2 kg1) | 5.40E-02 | | | | |
| SSA 870 nm | 9.98E-01 | | | | 9.96E-01 |
| SSA 405 nm | 9.74E-01 | 9.43E-01 | 9.41E-01 | 9.34E-01 | |
| AAE | 4.97E+00 | 6.85E+00 | 6.25E+00 | 7.24E+00 | |
| SSA 532 | | 9.90E-01 | 9.93E-01 | 9.91E-01 | |
| SSA 660 | | 9.93E-01 | 9.94E-01 | 9.91E-01 | |



`ef_sorted_by_property(chem, model_surrogate, property_variable).`
This function returns the individual NMOC_g EFs sorted by the specified property variable in ascending order. The NMOC_g is filtered by the specified fire type, chemical mechanism, and model surrogate.

**Table 13:** The output of `ef_sorted_by_property()` function when using the parameters- S22, XYNL, hc.

| mm | formula | compound | AVG_ temperate_ forest | N_ temperate_ forest | STD_ temperate_ forest | S22 | hc |
|---|---|---|---|---|---|---|---|
| 122 | C7H6O2 | Salicylaldehyde | 0.07 | 4 | 0.04 | XYNL | 6.00E-06 |
| 138 | C8H10O2 | Creosol | 0.3 | 7 | 0.19 | XYNL | 1.00E-06 |
| 124 | C7H8O2 | 2-methoxyphenol | 0.48 | 8 | 0.31 | XYNL | 1.00E-06 |
| 122 | C8H10O | 2,5-dimethyl phenol | 0.09 | 2 | 0.07 | XYNL | 1.00E-06 |
| 154 | C8H10O3 | Syringol | 0.08 | 7 | 0.07 | XYNL | 2.00E-07 |
| 110 | C6H6O2 | Resorcinol | 1.49 | 3 | 0.83 | XYNL | 1.00E-10 |

## 5.3 Query NMOC_g speciation profiles

The functions highlighted here are used for querying attributes of the NMOC_g speciation profiles.

`voc_profile(chem, fire_type).` This function returns the EF, moles, and mole fraction by model surrogate for the specified chemical mechanism and fire type.

**Table 14:** The output of `voc_profile ()` function when using the parameters- GEOSChem, peat.

| GEOSChem | $\sum$ EF | weighted_mm | $\sum$ moles | mole_fraction |
|---|---|---|---|---|
| PRPE | 1.25E+01 | 100.96 | 1.20E-01 | 1.30E-01 |
| MOH | 3.70E+00 | 31.00 | 1.20E-01 | 1.20E-01 |
| ACTA | 5.51E+00 | 60.00 | 9.00E-02 | 1.00E-01 |
| C2H6 | 2.53E+00 | 30.00 | 8.00E-02 | 9.00E-02 |
| ALK4 | 7.24E+00 | 105.08 | 7.00E-02 | 7.00E-02 |
| C3H8 | 4.42E+00 | 69.67 | 6.00E-02 | 7.00E-02 |
| C2H4 | 1.54E+00 | 28.00 | 5.00E-02 | 6.00E-02 |
| CH2O | 1.37E+00 | 29.50 | 5.00E-02 | 5.00E-02 |
| ALD2 | 1.82E+00 | 44.00 | 4.00E-02 | 4.00E-02 |
| CSL | 4.41E+00 | 130.55 | 3.00E-02 | 3.00E-02 |
| XYLE | 3.77E+00 | 121.06 | 3.00E-02 | 3.00E-02 |
| TOLU | 3.64E+00 | 120.76 | 3.00E-02 | 3.00E-02 |
| GLYC | 1.53E+00 | 60.00 | 3.00E-02 | 3.00E-02 |
| MVK | 1.99E+00 | 80.67 | 2.00E-02 | 3.00E-02 |
| BENZ | 1.33E+00 | 78.00 | 2.00E-02 | 2.00E-02 |
| HAC | 1.26E+00 | 74.00 | 2.00E-02 | 2.00E-02 |
| ACET | 9.40E-01 | 58.00 | 2.00E-02 | 2.00E-02 |
| OCS | 7.30E-01 | 60.00 | 1.00E-02 | 1.00E-02 |
| MEK | 1.16E+00 | 107.89 | 1.00E-02 | 1.00E-02 |
| ISOP | 6.60E-01 | 68.00 | 1.00E-02 | 1.00E-02 |
| HCOOH | 4.20E-01 | 46.00 | 9.00E-03 | 1.00E-02 |
| EOH | 3.50E-01 | 46.00 | 8.00E-03 | 8.00E-03 |
| MGLY | 4.80E-01 | 79.00 | 6.00E-03 | 6.00E-03 |
| MACR | 3.60E-01 | 63.00 | 6.00E-03 | 6.00E-03 |
| RCHO | 5.10E-01 | 101.40 | 5.00E-03 | 5.00E-03 |



| BALD | 4.60E-01 | 128.00 | 4.00E-03 | 4.00E-03 |
| MTPA | 2.50E-01 | 136.00 | 2.00E-03 | 2.00E-03 |
| R4N2 | 1.20E-01 | 114.50 | 1.00E-03 | 1.00E-03 |
| NAP | 1.20E-01 | 128.00 | 9.00E-04 | 1.00E-03 |
| MTPO | 1.70E-01 | 192.67 | 9.00E-04 | 9.00E-04 |
| PYAC | 6.00E-02 | 88.00 | 7.00E-04 | 7.00E-04 |
| DMS | 4.00E-02 | 62.00 | 6.00E-04 | 7.00E-04 |
| CH3Br | 3.00E-02 | 94.00 | 3.00E-04 | 3.00E-04 |
| CH3I | 2.00E-02 | 141.00 | 2.00E-04 | 2.00E-04 |
| MP | 2.00E-02 | 118.00 | 2.00E-04 | 2.00E-04 |

`weighted_property( fire_type, chem).` This function calculates the EF-weighted molar mass (mm), OH rate constant (kOH), logarithm of saturation concentration (cstar), and vapor pressure (vp) for the specified chemical mechanism and fire type.


**Table 15:** The output of `weighted_property()` function when using the parameters- boreal forest, MOZART-T1.

| MOZT1 | mm | kOH | cstar | vp |
|---|---|---|---|---|
| BIGENE | 75.64 | 6.00E-11 | 9.09 | 1067.17 |
| CH3OH | 31.00 | 9.00E-13 | 8.88 | 127.00 |
| C2H4 | 28.21 | 8.00E-12 | 10.34 | 50000.00 |
| CH3COOH | 60.00 | 7.00E-13 | 7.64 | 15.70 |
| CH2O | 30.00 | 8.00E-12 | 9.78 | 3890.00 |
| NROG | 57.06 | 2.00E-12 | 9.87 | 1231.96 |
| TOLUENE | 92.62 | 3.00E-11 | 7.56 | 100.42 |
| CH3CHO | 44.00 | 1.00E-11 | 9.55 | 902.00 |
| C2H6 | 30.00 | 2.00E-13 | 10.61 | 30000.00 |
| C3H6 | 42.06 | 3.00E-11 | 10.06 | 8543.36 |
| XYLENES | 111.96 | 6.00E-11 | 7.24 | 33.77 |
| MEK | 95.92 | 2.00E-11 | 7.54 | 23.71 |
| PHENOL | 95.47 | 3.00E-11 | 6.81 | 0.34 |
| C2H2 | 26.59 | 1.00E-12 | 10.27 | 40000.00 |
| CH3COCH3 | 58.01 | 2.00E-13 | 9.06 | 229.83 |
| GLYALD | 60.00 | 1.00E-11 | 7.40 | 0.91 |
| BIGALK | 99.90 | 3.00E-11 | 8.43 | 341.32 |
| BENZENE | 78.03 | 1.00E-12 | 8.35 | 92.82 |
| HCOOH | 46.00 | 5.00E-13 | 7.79 | 42.60 |
| BPIN | 136.00 | 6.00E-11 | 7.49 | 2.72 |
| C3H8 | 44.53 | 3.00E-12 | 10.06 | 6953.16 |
| CH3COCHO | 72.00 | 1.00E-11 | 8.64 | 121.00 |
| CRESOL | 128.67 | 6.00E-11 | 5.83 | 0.09 |
| ISOP | 68.00 | 1.00E-10 | 8.99 | 550.00 |
| APIN | 136.00 | 7.00E-11 | 7.48 | 4.11 |
| MTERP | 131.49 | 2.00E-10 | 7.71 | 3.72 |
| HYAC | 74.00 | 3.00E-12 | 7.09 | 1.74 |
| MVK | 70.00 | 2.00E-11 | 8.47 | 91.30 |
| BZALD | 115.48 | 2.00E-11 | 6.04 | 0.83 |
| LIMON | 136.00 | 2.00E-10 | 8.47 | 1.30 |
| MYRC | 136.00 | 2.00E-10 | 7.17 | 2.18 |
| MACR | 70.00 | 3.00E-11 | 8.71 | 155.00 |
| C2H5OH | 46.00 | 3.00E-12 | 8.63 | 59.30 |
| BCARY | 203.83 | 2.00E-10 | 6.01 | 0.03 |
| TERPROD1 | 196.00 | 1.00E-11 | 5.93 | 0.23 |
| ALKNIT | 109.04 | 7.00E-13 | 7.99 | 46.57 |
| BIGENE | 75.64 | 6.00E-11 | 9.09 | 1067.17 |




## 6. Conclusions

NEIVA represents the most comprehensive EF compilation for globally-relevant fuel types, and uniquely includes selected laboratory data. NEIVA was created by integrating EF data from Akagi et al. (2011) and 30 papers published since the 2014 and 2015 updates to Akagi. The most significant expansion of data occurred for temperate forest, peat, and crop residue fires. EF data are stored in several datasets that represent varying levels of data processing, merging, and

averaging. All datasets can be accessed through the NEIVA GitHub site. NEIVA has been structured so that new EF data can easily be added and recommended averages recalculated. EF data can be flexibly queried with varying levels of detail from the individual study level to averaged across all studies for a given fuel or fire type, and from the individual compound or constituent level to representative model surrogate species. In addition, NEIVA has been

structured to enable efficient inclusion of EF data into chemical mechanisms allowing for better attribution of biomass burning emissions and impacts in future model studies.

Inclusion of adjusted laboratory data increases the number of data points and number of compounds represented without introducing variability or uncertainty outside of what is expected

and what has been observed in field studies. The number of NMOC_g represented in NEIVA is up to an order of magnitude higher than in the most recent EF compilations. Inclusion of this more diverse set of NMOC_g changes property distributions that can affect predictions of atmospheric composition and chemistry, illustrated here using volatility and OHR. Further, mapping this more diverse set of NMOC_g to model surrogates leads to distinct differences in the surrogate

distributions when compared with other existing compilations that are likely to affect multiscale model predictions. NEIVA has a better representation of IVOCs, resulting in a shift in the volatility distribution to lower volatilities, with the lowest volatility bin shifted by up to three orders of magnitude. In addition, the NEIVA NMOC_g speciation profiles when mapped to SAPRC-07 model surrogates resulted in higher OHR by 40-90%, which likely is conservative since the $k_{OH}$

values were not updated to represent measured compound distributions and the greater NMOC_g/CO ratio for some fuel types was not considered.



**Code and Data Availability**

The NEIVA datasets (SQL and CSV formats), Python script files used to generate the datasets, and Jupyter notebooks with instructions for adding new data and examples for querying the datasets and are freely available on GitHub (https://github.com/NEIVA-BB-Emissions-Inventory/NEIVAv1.0; last accessed February 2024). The datasets are also permanently archived
on Zenodo via Binte Shahid et al. (2024) with the link https://doi.org/10.5281/zenodo.10721105 under the GNU General Public License version 2.0 or later.

**Author Contributions**

     K. C. B, C. W., and R. J. Y. conceived of the idea for a more flexible and sustainable emission
factor database. C. W. and R. J. Y. provided input throughout the development of the database. S. B.-S. led development of the database, including writing associated Python script files and functions and devising methodologies for assigning unique identifiers, merging datasets, and mapping to model surrogates. S. B.-S. generated all figures for the manuscript. S. B.-S. developed the GitHub site and set up code and data archiving. F. G. L. provided input on the
GitHub and Python scripts and model use-cases. K. C. B. led writing of the manuscript, with significant input from S. B.-S. K. C. B. and S. B.-S. led writing of the SI. All authors contributed to editing the manuscript and SI.

**Competing interests**

The contact author has declared that none of the authors has any competing interests.

**Acknowledgements**

     K. C. B. and S. B.-S. were supported by NOAA AC4 grant NA17OAR4310007 and NSF CAREER grant AGS-1753364. R. J. Y. was supported by NSF grant AGS-1748266. F. G. L.
was supported by a collaborative MOU with the CDC.



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
