# Peer review of "NEIVAv1.0: Next-generation Emissions InVentory expansion of Akagi et al. version 1.0"

_EGUsphere, 2024_

## Author Comment (AC1)

**Response to Review #1**

We thank the referees for their careful review and constructive comments. We made minor corrections to the manuscript based on the referee comments. Below please find our responses to these comments (in blue).

NIEVA will be a great benefit to researchers concerned with the effects of biomass burning on atmospheric composition, air quality, climate, and public health, especially those working at continental to global scales.

We are pleased to hear that the paper has been well-received and thankful that the referee recognizes the importance of this newly developed database.

1. The GitHub repository does not include ".csv" tables as described in the manuscript; however, I was able to create and download using the provided tools (in Google Colab)

The CSV files of Integrated EF, Processed EF and Recommended EF have now been added to the GitHub repository.

2. The inclusion of Travis et al. (2023) for prescribed burns of shrublands in the central U.S. with chaparral seems a bit odd. Did the authors consider introducing a separate shrubland category?

We agree. Globally, shrublands are diverse and are under-sampled. So while it is true that chaparral has differences with central US shrublands such as mesquite or sagebrush in the west, for now we are doing our best to fill in this broad category rather than create more categories with little supporting data. Having said that, a strength of NEIVA is the data for individual fuel types can be extracted.

3. L346-348: "Rice straw EFs measured during a FIREX laboratory pile-burning simulation also were included (Koss et al.,2018; Selimovic et al., 2018; Gkatzelis et al., 2023; Travis et al., 2023)." Travis et al. and Gkatzelis et al. are both field studies and the latter does not report crop residue emissions. The sentence or references need updating.

Thank you for catching this. The reference has been corrected and now reads:
Rice straw EFs measured during a FIREX laboratory pile-burning simulation also were included (Koss et al.,2018; Selimovic et al., 2018)."

4. Table S14. Travis et al. (2023) is a field study, the eastern portion of FIREX-AQ 2019. However, Table S14 lists Travis et al. (2023) as laboratory measurements and indicates the study's EFs were adjusted in the creation of the NIEVA Processed EF dataset. The intermediate datasets (".csv") I extracted indicate the Travis et al. was correctly processed as field data suggesting the table is in error. However, I did not re-run the processing scripts to verify this is the case. This should be confirmed.

Yes, the Travis et al., 2023 study was included as a field study in the data processing steps. Table S14 has been corrected to reflect Travis et al. is a field study and not a lab study.

5. L507-508: "For boreal forest, the relatively high laboratory-based CO value is largely driven by EFs measured in boreal peat studies and reported by Yokelson et al. (1997)." Lab measurements of boreal peat should be included with peat. Is this inclusion in boreal forest fires a legacy of Akagi et al. (2011)? May have been better to extract subset of studies from Akagi et al (2011) instead of using full dataset if one lab burn has such a big impact.

A brief background may be helpful here. Akagi et al., 2011 calculated a "peatland" category assuming the overstory for peat fires was always evergreen tropical "peat swamp" forest, following Page et al. (2002). However, it is now clear that tropical peat fires are mostly human-caused and involve various overstories, including crops, forests, and more. Thus, in the tropics, it is better to calculate peat-fire emissions and surface-fire emissions as separate additive components from the same area. Note that peat underlies only a small percentage of tropical forests, as shown in the global peat map (https://peatlands.org/peatlands/where-can-peatlands-be-found/). In contrast, peat is a widespread component of boreal forests, where fires are mostly a natural disturbance in the forest life cycle. Boreal forest fires often have an intense flaming crown-fire phase followed by protracted smoldering of surface and belowground fuels.

For the boreal forest fire category, measurements of NMOG emissions from common boreal ground-level and belowground fuels are mostly lab-based, while field studies are airborne, potentially underestimating smoldering emissions. Our peat category already includes the Yokelson et al., 1997 and many other peat fire studies. In our boreal lab category, we include lab studies of NMOG emissions from boreal fuels that tend to burn more by flaming (e.g., black spruce canopy) and those that tend to burn more by smoldering (e.g., peat, duff, organic soils, woody dead/down materials). Dropping the one peat value from our current lab average would only decrease the lab average EF_CO by about 10%, so we incorrectly overstated the impact of the peat value. We have revised the unintentionally misleading sentence (see below). Retaining some representation of boreal peat fire NMOG emissions makes our lab category average more representative of the overall ecosystem. Thus, we have not deleted the peat study or recalculated the average lab EF_CO.

It is likely that our current average EFs, including all adjusted lab data, will better represent the NMOG chemistry of aircraft-measured or remotely-sensed smoke subject to immediate lofting and long-range transport. Given the remote location, long lifetime, and multiple injection altitudes, it is exceptionally difficult to estimate the best weighting of studies to represent the total "flaming to smoldering" ratio from boreal fires, including prolonged residual smoldering. We have opted to use a consistent lab/field approach across fuel types. However, the database structure intentionally facilitates other choices, and we can refer the reader to a more extensive discussion on boreal forest fire average MCE or EFCO elsewhere (Akagi et al., 2011; Wiggins et al., 2021).

Old: "For boreal forest, the relatively high laboratory-based CO value is largely driven by EFs measured in boreal peat studies and reported by Yokelson et al. (1997)."

New: "For boreal forest, the laboratory-based EFCO value is about 20% higher than the field average, which is based exclusively on airborne studies. More detailed discussion of averaging studies for this fire type can be found elsewhere (Akagi et al., 2011, Wiggins et al., 2021)."

6. A peat emissions field study worthy of consideration for future updates: Geron & Hays (2013) Air emissions from organic soil burning on the coastal plain of North Carolina, Atmospheric Environment, Volume 64, 2013, Pages 192-199, ISSN 1352-2310,https://doi.org/10.1016/j.atmosenv.2012.09.065.

Thank you for the suggestion. This paper, along with future papers and potentially modified averaging schemes, can be implemented by the users or by 'us' as part of future updates to the main NEIVA database.

---

## Author Comment (AC3)

**Response to Review #2**

We thank the referee for the careful review and constructive comments. We made minor corrections to the manuscript based on the referee comments. Below please find our responses to these comments (in blue).

The methodology is well documented and the differences with other inventories are thoroughly discussed. The discussion on applications and impacts in modeling is a good addition. The manuscript is very well written and easy to follow.

We are pleased to hear that the paper is well-received.

1. The PM category "PM2.5* = PM1-PM5" is not very clearly defined. Is it PM between 1μm and 5μm? Why report this particular category, which is not very common? I think some clarification is needed here.

Fresh BB PM emissions typically exhibit a bimodal size distribution with peaks near 0.3 and 10 microns in diameter and a valley from 1 to 5 microns (Figure 1). Cyclones or impactors are often used to select particles below/above a nominal aerodynamic diameter based on their inertia in a flow, but the transmission curve is sigmoidal, with 0-100% transmission typically occurring over a span of approximately 2 microns (Figure 2). The 50% cutoff is used as the nominal aerodynamic diameter that is 'selected' by a cyclone, for example, when used under specific conditions. In practice, especially in airborne use, the conditions (such as flow rate) may vary, shifting the 50% cutoff diameter higher or lower, e.g., with slower/faster flow. This is sometimes measured in experiments, which then report, for example, 'PM3.5.' For our purposes, since PM1 typically accounts for about 80% of PM2.5, and 2.5 microns is in an extended valley in the size distribution, we have binned reported EFs for PM1 through PM5 to build statistics for what is mostly fine, mostly organic particulate matter. We maintain separate categories for PM10 and TSP, which, in contrast, may contain more entrained dust, debris, and biological fragments.

[Figure]

Figure 1:  Retrieved AERONET particle size distributions for a 440 nm Aerosol Optical Depth (AOT) of 0.7. (Reference-Reid, J. S., Koppmann, R., Eck, T. F., and Eleuterio, D. P.: A review of biomass burning emissions part II: intensive physical properties of biomass burning particles, Atmos. Chem. Phys., 5, 799–825, doi:10.5194/acp-5-799-2005, 2005.)

[Figure]

Figure 2: PM transmission sigmoidal curve from TSI.

We have added the following sentence to the main manuscript to improve clarity:

$PM_{2.5}^{*}$ subcategory accounts for the fact that fine or accumulation mode PM may be reported at multiple size cuts (e.g., $PM_1$, $PM_{3.5}$) based on instrument specifications and operating conditions.

2. Section 5: the manuscript is generally quite long and I do not think Section 5 helps the manuscript, especially since the GitHub is well commented and documented. I would take out Section 5 or move it to supplementary material.

We agree with this reviewer regarding the length of Section 5 and duplication with information on GitHub. We also think that the tables themselves highlight contents of the database that were not described in any detail in the main manuscript. In response to this comment, we have moved this section to an appendix, so that it stays with the manuscript but not in line with the more important text.

3. Supplementary material: some references need fixing ("Error! Reference source not found." several times)

The SI Table references have been updated and the "error" appearances have been corrected in the SI.